# Probabilistic Robustness Analysis in High Dimensional Space: Application to Semantic Segmentation Networks

## Abstract

Semantic segmentation networks (SSNs) are central to safety-critical applications such as medical imaging and autonomous driving, where robustness under uncertainty is essential. However, existing probabilistic verification methods often fail to scale with the complexity and dimensionality of modern segmentation tasks, producing guarantees that are overly conservative and of limited practical value. We propose a probabilistic verification framework that is architecture-agnostic and scalable to high-dimensional input-output spaces. Our approach employs conformal inference (CI), enhanced by a novel technique that we call the **clipping block**, to provide provable guarantees while mitigating the excessive conservatism of prior methods. Experiments on large-scale segmentation models across CamVid, OCTA-500, Lung Segmentation, and Cityscapes demonstrate that our framework delivers reliable safety guarantees while substantially reducing conservatism compared to state-of-the-art approaches on segmentation tasks.

## 1 Introduction

Semantic image segmentation is widely applied in high-stakes areas, including medical imaging (Perone et al., 2018) and autonomous driving (Deng et al., 2017). Yet, segmentation models powered by deep learning are known to be fragile, as they can be easily manipulated by adversarial inputs (Xie et al., 2017), limiting their safe use in these critical applications.

Achieving certified robustness in segmentation is particularly difficult, since every individual element (such as a pixel in an image) must be verified at once. The scale of datasets and models in this setting often exceeds the capacity of existing deterministic verification techniques (Zhou et al., 2024; Tran et al., 2020b), while probabilistic approaches (Cohen et al., 2019; Jeary et al., 2024) must contend with the compounded uncertainty introduced by the vast number of per-element certifications. This compounded uncertainty can be formalized using the union bound[1], which weakens the probabilistic guarantees of these verification algorithms as the number of pixels grows—explaining why they are generally limited to classification tasks and do not scale to segmentation tasks.

A prominent research direction in probabilistic verification is randomized smoothing, first introduced for classifiers Cohen et al. (2019) and later adapted to segmentation models Fischer et al. (2021), with subsequent improvements in Anani et al. (2024); Hao et al. (2022). More recently, conformal inference has been investigated as a means to provide probabilistic guarantees in classification tasks Jeary et al. (2024). In Hashemi et al. (2025), the authors proposed a methodology to extend the application of conformal inference from robustness analysis in classification task to robustness analysis in segmentation tasks. They first introduce a naive approach that relies solely on conformal inference, which produces only hypercubes for the reachable set of the vector of logits—also described in the preliminaries of this paper. They then improve this method by training a small ReLU neural network as a surrogate for the base segmentation model, demonstrating that the surrogate enables the construction of more general convex shapes for the system's reachable set.

Appendix A presents a comprehensive comparison between segmentation verification techniques based on randomized smoothing Fischer et al. (2021); Anani et al. (2024) and the conformal in-

---

[1] $\Pr\left[P_1 \wedge P_2 \wedge \ldots \wedge P_n\right] \geq 1 - \sum_{i=1}^{n}\left(1 - \Pr[P_i]\right)$

ference approach in Hashemi et al. (2025), highlighting the advantages of conformal inference for segmentation tasks. However, as we detail later, the method in Hashemi et al. (2025) has its own limitations, which we address in this paper. In fact, the small ReLU neural network introduced in Hashemi et al. (2025) poses several challenges for this methodology. First, the surrogate model can suffer from fidelity issues, leading to overly conservative guarantees. Second, the high dimensionality of the input space makes training infeasible. Moreover, it imposes restrictive assumptions on the perturbation of the image, requiring it to be sparse and confined to $\ell_\infty$-type perturbations.

In this paper, we propose an alternative technique to replace this ReLU surrogate model and to avoids all these issues. Specifically, we introduce a **clipping-block** at the end of the base model. This approach requires no surrogate training—eliminating fidelity concerns—and allows perturbations across the entire image, supporting general $\ell_p$ perturbations. To show this numerically, we deliberately consider the same models and datasets as in Hashemi et al. (2025), but with higher-dimensional (and also full) image perturbations in our numerical experiments and we observed that the surrogate-based approach in Hashemi et al. (2025) fails under these conditions because deterministic reachability using the NNV toolbox Tran et al. (2020b) cannot handle the dimensionality of the input sets used in our experiments facing runtime and out-of-memory errors.

**Related Works**.

**Deterministic Verification of Neural Networks**. Deterministic verification of neural networks has been addressed using several well-established techniques, such as Satisfiability Modulo Theories (SMT) solvers (Duong et al., 2023; Katz et al., 2017; Wu et al., 2024), Mixed Integer Linear Programming (MILP) formulations (Anderson et al., 2020; Cheng et al., 2017; Tjeng et al., 2017), and different types of reachability analysis (Bak et al., 2020; Tran et al., 2020a). Many of the existing methods can be also viewed through the general framework of branch-and-bound algorithms (Bunel et al., 2020), which iteratively partition and prune the input domain to either uncover counterexamples or prove correctness. Prominent tools in this area are, $\alpha, \beta$-CROWN, (Zhou et al., 2024), PyRAT (Lemesle et al., 2024), and Marabou (Wu et al., 2024). Existing deterministic techniques do not scale to segmentation models because of their high level of complexity and dimensionality.

**Conformal Inference**. The integration of CI with formal verification techniques has recently received noticeable interest, that is primarily due to its accuracy and level of scalability. For instance, Bortolussi et al. (2019) merges CI with neural state classifiers to develop a stochastic runtime verification algorithm. Lindemann et al. (2023); Zecchin et al. (2024) employs CI to guarantee results in MPC control using a trained model. Tonkens et al. (2023) applies CI for planning with probabilistic safety guarantees, and Hashemi et al. (2023; 2024) employs CI for reachability of stochastic dynamical systems, see Lindemann et al. (2024) for a recent survey article.

## 2 PRELIMINARIES

**Notations**. We write $x \sim \mathcal{X}$ to indicate that the random variable $x$ is drawn from the distribution $\mathcal{X}$, and $x \overset{\mathcal{X}}{\sim} \mathbf{X}$ to indicate that $x$ is sampled from the set $\mathbf{X}$ according to the distribution $\mathcal{X}$. We use betacdf to denote the CDF of beta distribution. We use bold letters to represent sets and calligraphic letters to denote distributions. Consider a 3-dimensional tensor $x \in \mathbb{R}^{h \times w \times nc}$ of height $h$, width $w$, and $nc$ channels, which we flatten into a vector. The vectorized form is denoted as $\mathbf{vec}(x) \in \mathbb{R}^{n_0}$, where $n_0 = nc \times w \times h$. The Minkowski sum is denoted by $\oplus$. The ceiling operator $\lceil x \rceil$ denotes the smallest integer greater than or equal to $x \in \mathbb{R}$. Here, we denote a $\ell_2$ ball with radius $r \in \mathbb{R}^{n_0}$ and center $x \in \mathbb{R}^{n_0}$ with $\mathbf{B}_r(x)$.

**Semantic Segmentation Neural Networks (SSN)** Consider an input image $x \in \mathbb{R}^{h \times w \times nc}$. An SSN processes this image and outputs a 2-D label mask, assigning a class to every pixel location $(i, j) \in \{1, \ldots, h\} \times \{1, \ldots, w\}$, we have: $\text{SSN}(x) = \text{mask} \in \mathbf{L}^{h \times w}, \quad \mathbf{L} = \{1, 2, \ldots, L\}$. The network computes the logits $y \in \mathbb{R}^{h \times w \times L}$ in the last layer. For each pixel, the predicted label is obtained by choosing the class with the largest logit value: $\text{mask}(i, j) = \arg\max_{\ell \in L} y(i, j, \ell)$.

Instead of working directly with multidimensional tensors, it is sometimes convenient to flatten both the input image and the logits. Let $n_0 = h \cdot w \cdot nc$ and $n = h \cdot w \cdot L$. Then the map between flattened

input $\mathbf{vec}(x)$ and the flattened logit values $\mathbf{vec}(y)$ can equivalently be viewed as a nonlinear function

$$f : \mathbb{R}^{n_0} \to \mathbb{R}^n, \quad f(\mathbf{vec}(x)) = \mathbf{vec}(y),$$

**Conformal Inference and $\langle \epsilon, \ell, m \rangle$ guarantee** Suppose we have $m$ i.i.d. non-negative random variables $\mathbf{M} = \{R_1, R_2, \ldots, R_m\}, \quad R_1 < R_2 < \cdots < R_m$, drawn from some distribution $R \sim \mathcal{D}$. Conformal inference (CI) (Vovk et al., 2005) provides a way to construct prediction intervals with guaranteed coverage at a user-specified miscoverage level $\epsilon \in (0, 1)$.

For a new draw $R_{m+1}$ from the same distribution $R_{m+1} \sim \mathcal{D}$, CI defines a prediction interval $C(R_{m+1}) = [0, d]$ such that $\Pr \left[ R_{m+1} \in C(R_{m+1}) \right] \geq 1 - \epsilon$.

In this setting, the values $R_i \in \mathbf{M}$ are often called **nonconformity scores**, and the set $\mathbf{M}$ is referred to as the calibration dataset. To compute the threshold $d$, one considers the empirical distribution of these scores. Specifically, define $\ell = \lceil (m + 1)(1 - \epsilon) \rceil$, and let $R_\ell$ denote the $\ell$-th smallest element ($\ell < m$) of the set $\mathbf{M}$ (Tibshirani et al., 2019). Then the prediction threshold is given by $d = R_\ell$, and we are guaranteed the following marginal coverage guarantee:

$$\Pr \left[ R_{m+1} \leq R_\ell \right] \geq 1 - \epsilon, \tag{1}$$

Here, the probability is marginal which means that it is taken jointly over both the randomness of calibration dataset $\mathbf{M}$ and the test sample $R_{m+1}$ (Tibshirani et al., 2019; Vovk et al., 2005). The test sample $R_{m+1}$ is assumed to be drawn from the same underlying distribution as the calibration set. Following the convention in this paper, we call this new sample **unseen** and denote it by $R^{\text{unseen}}$. Since we do not assume randomness in the calibration dataset, following the proposition of Vovk et al. (2005) the authors in Hashemi et al. (2025) use a two-step representation of the guarantee:

$$\Pr \left[ \ \Pr \left[ \ R^{\text{unseen}} \leq R_\ell \ \right] > 1 - \epsilon \ \right] > 1 - \mathsf{betacdf}_{1-\epsilon}(\ell, m + 1 - \ell), \tag{2}$$

where the outer probability captures the randomness in $\mathbf{M}$, while the inner probability is valid for a fixed set $\mathbf{M}$. For further details and clarifying examples, please see Hashemi et al. (2025).

Here, we tune $m, \ell$, such that for a given $\epsilon \in [0, 1]$, $\mathsf{betacdf}_{1-\epsilon}(\ell, m + 1 - \ell)$ becomes less than a user-specified threshold. Henceforth, for any logical statement $P(\ell, m) \in \{\text{true}, \text{false}\}$, defined over the hyper-parameters $m, \ell \leq m$, we refer to the double-step guarantee,

$$\Pr \left[ \ \Pr \left[ \ P(\ell, m) \ \right] > 1 - \epsilon \ \right] > 1 - \mathsf{betacdf}_{1-\epsilon}(\ell, m + 1 - \ell)$$

as the $\langle \epsilon, \ell, m \rangle$ guarantee and we say $P(\ell, m)$ satisfies a $\langle \epsilon, \ell, m \rangle$ guarantee. We refer to $\delta_1 := 1 - \epsilon$, as the coverage level and $\delta_2 := 1 - \mathsf{betacdf}_{1-\epsilon}(\ell, m + 1 - \ell)$ as the confidence level.

**Deterministic and Probabilistic Reachability Analysis on Neural Networks** Reachability analysis for a neural network $f : \mathbb{R}^{n_0} \to \mathbb{R}^n$ has been approached in the literature in two ways. In *deterministic reachability analysis*, given an input set $\mathbf{I} \subset \mathbb{R}^{n_0}$, the goal is to construct a set $\mathbf{R}_f(\mathbf{I}) \subset \mathbb{R}^n$ that contains all possible outputs:

$$x \in \mathbf{I} \quad \Rightarrow \quad f(x) \in \mathbf{R}_f(\mathbf{I}). \tag{3}$$

*Probabilistic reachability analysis*, in contrast, assumes a prior distribution $\mathcal{W}$ for inputs over $\mathbf{I}$, denoted $x \overset{\mathcal{W}}{\sim} \mathbf{I}$. For a given miscoverage level $\epsilon$, it aims to construct a set $\mathbf{R}_f^\epsilon(\mathcal{W}) \subset \mathbb{R}^n$ such that the network output lies within this set with high probability:

$$x \overset{\mathcal{W}}{\sim} \mathbf{I} \quad \Rightarrow \quad \Pr \left[ f(x) \in \mathbf{R}_f^\epsilon(\mathcal{W}) \right] \geq 1 - \epsilon. \tag{4}$$

In this work, we detail a procedure to compute such a probabilistic reach set for a given miscoverage level $\epsilon$ and sampling distribution $x \overset{\mathcal{W}}{\sim} \mathbf{I}$. This involves constructing a calibration set $\mathbf{M}$ of size $m$ and selecting a rank $\ell$ to enforce the desired coverage. To maintain consistent terminology, we express the probabilistic reach set using the $\langle \epsilon, \ell, m \rangle$ guarantee and denote it as $\mathbf{R}_f^\epsilon(\mathcal{W}\,;\ell, m)$, which satisfies

$$x \overset{\mathcal{W}}{\sim} \mathbf{I} \quad \Rightarrow \quad \Pr \left[ \Pr \left[ f(x) \in \mathbf{R}_f^\epsilon(\mathcal{W}\,;\ell, m) \right] \geq 1 - \epsilon \right] \geq 1 - \mathsf{betacdf}_{1-\epsilon}(\ell, m + 1 - \ell).$$

**Adversarial Examples and Robustness of SSNs**   Adversarial attack is a perturbation on an image by adding a combination of noise images $x_1^{\text{noise}}, \ldots, x_r^{\text{noise}}$ weighted by a vector of coefficients $\lambda = [\lambda(1), \ldots, \lambda(r)]^\top$. Formally, an adversarial image is generated via

$$x^{\text{adv}} = \Delta_{\lambda, x^{\text{noise}}}(x) = x + \sum_{i=1}^r \lambda(i) x_i^{\text{noise}}, \quad \lambda \in [\underline{\lambda}, \bar{\lambda}] \subset \mathbb{R}^r, \tag{5}$$

where the coefficients are unknown but bounded, defining the set of feasible adversarial inputs $x^{\text{adv}} \in \mathbf{I}$. The impact of these perturbations on semantic segmentation networks (SSNs) can be analyzed at the pixel level.

The SSN at pixel (i,j) is **robust** if its predicted label is invariant under all allowed perturbations: $\mathsf{SSN}(\Delta_{\lambda, x^{\text{noise}}}(x))(i, j) = \mathsf{SSN}(x)(i, j), \quad \forall \lambda \in [\underline{\lambda}, \bar{\lambda}]$. If the label changes under all perturbations, the pixel is **non-robust**: $\mathsf{SSN}(\Delta_{\lambda, x^{\text{noise}}}(x))(i, j) \neq \mathsf{SSN}(x)(i, j), \quad \forall \lambda \in [\underline{\lambda}, \bar{\lambda}]$. A pixel is classified as **unknown** when some perturbations change its label while others do not.

To quantify robustness across the entire image, we utilize a metric known as *Robustness Value (RV)*, defined as the percentage of pixels that remain robust under adversarial attacks: $RV = 100 \times (N_{\text{robust}}/N_{\text{pixels}})$, where $N_{\text{pixels}} = h \times w$. This framework allows assessing SSN performance under adversarial conditions, highlighting which regions of an image are consistently robust, non-robust, or uncertain.

Once the reachset $\mathbf{R}_f^\epsilon(\mathcal{W}; \ell, m)$ over the SSN logits $y \in \mathbb{R}^n$ is obtained, we can analyze each pixel status by projecting it onto individual logit components. This projection yields, for every pixel location $(i, j) \in \{1, \ldots, h\} \times \{1, \ldots, w\}$, a set of $L$ intervals corresponding to the class labels $l \in \mathbf{L}$.

For pixel $(i, j)$, let $l^*(i, j) \in \mathbf{L}$ denote the class whose logit interval has the largest lower bound. The pixel is classified as *unknown* if this lower bound does not strictly exceed the upper bounds of all other class intervals, indicating that multiple labels are possible under perturbation. Otherwise, if the lower bound of $l^*(i, j)$ is strictly greater than all others, the pixel is labeled based on the original prediction: it is *robust* if $l^*(i, j) = \mathsf{SSN}(x)(i, j)$ and *non-robust* if $l^*(i, j) \neq \mathsf{SSN}(x)(i, j)$. Algorithm 2 provides a detailed step-by-step procedure for computing the pixel-wise status using this approach.

**Problem Formulation**   Given a SSN, $x \mapsto \mathsf{SSN}(x)$, we address two problems:

**Problem 1**. Given an input image $x$, and a desired miscoverage level $\epsilon$, the goal is to determine whether each pixel $x(i, j)$ is robust, non-robust or unknown to a set of adversarial examples $x^{\text{adv}} \in \mathbf{I}$ with strong probabilistic guarantees.

**Problem 2**. Given $K$ test images $\{x_1, \ldots, x_K\}$, and a set of adversarial examples $\mathbf{I}$, considering a miscoverage level $\epsilon$, compute the average robustness value $\overline{RV}$, with strong probabilistic guarantees.

## 3   SCALING PROBABILISTIC REACHABILITY ANALYSIS ON SSN WITH STRONG GUARANTEES USING CLIPPING-BLOCK

As the architecture of a deep neural network becomes more complex, the distribution of its output can exhibit increasingly intricate behavior. Appendix B summarizes the naive technique proposed by Hashemi et al. (2025) for the reachability of SSNs. As it is explained in Appendix B, this technique for reachability via conformal inference remains robust to any output distribution that represents well a sampled calibration dataset, allowing it to handle such complexities reliably. However, the tightness of this reachset does not persist when the dimensionality of the network's output increases.

- The first issue is that the naive technique can only produce a hyper-rectangular reachset, that can be a significant source of conservative in high-dimensional spaces.
- The second issue is that the space of distributions $\mathcal{Y}$ that can provide some distribution $\mathcal{D}$ which well represent a sampled calibration dataset is significantly larger in high dimensional space, resulting in a conservative reachable set.

To tackle these challenges, Hashemi et al. (2025) propose constructing a more flexible reachset in the form of a general convex hull, rather than a simple hyper-rectangle. This is achieved by

approximating the original neural network $f$ with an alternative computation graph $g : \mathbb{R}^{n_0} \to \mathbb{R}^n$, for which a guaranteed deterministic reachable set can be derived. To obtain the surrogate model $g$ they suggest training a small ReLU neural network over the train dataset $\mathbf{T}$ and then compute its deterministic reachset over $\mathbf{I}$ using NNV toolbox Tran et al. (2020b). Since $g$ is is trained on $\mathbf{T}$, we infere it is accurate only for inputs $x$ sampled from the set $\mathbf{I}$. The core idea is to utilize its deterministic reachable set (also known as surrogate reachset), and then **inflate** it to **account for approximation error** between $f$ and its surrogate $g$, thereby obtaining a probabilistic reachset for the original model $f$ with an $\langle \epsilon, \ell, m \rangle$ guarantee.

To inflate the surrogate reachset, Hashemi et al. (2025) adopt the procedure from the naive approach to construct a hyper-rectangle that bounds the prediction error while maintaining a provable $\langle \epsilon, \ell, m \rangle$ guarantee. Concretely, let $q(\mathbf{vec}(x)) = f(\mathbf{vec}(x)) - g(\mathbf{vec}(x))$. The naive technique is then applied to $q(\mathbf{vec}(x))$ to compute its hyper-rectangular reachset under the same $\langle \epsilon, \ell, m \rangle$ guarantee. This resulting hyper-rectangle is referred to as the *inflating set*. In conclusion, if we denote the deterministic reachset of surrogate model as $\mathcal{R}_g(\mathbf{I})$,

$$x \in \mathbf{I} \;\;\Rightarrow\;\; g(\mathbf{vec}(x)) \in \mathcal{R}_g(\mathbf{I})$$

and for some hyper-parameters $(\ell, m)$ and distribution $x \overset{\mathcal{W}}{\sim} \mathbf{I}$, the $\langle \epsilon, \ell, m \rangle$ guaranteed probabilistic reachset for prediction errors by $\mathcal{R}_q^\epsilon(\mathcal{W}; \ell, m)$,

$$x \overset{\mathcal{W}}{\sim} \mathbf{I} \;\Rightarrow\; \Pr \big[\; \Pr[\; q(\mathbf{vec}(x)) \in \mathcal{R}_q^\epsilon(\mathcal{W}; \ell, m) \;] > 1 - \epsilon \;\big] > 1 - \mathsf{betacdf}_{1-\epsilon}(\ell, m + 1 - \ell),$$

then the probabilistic reachset of the model $f$ with the same $\langle \epsilon, \ell, m \rangle$ guarantee is obtainable by $\mathcal{R}_f^\epsilon(\mathcal{W}; \ell, m) = \mathcal{R}_g(\mathbf{I}) \oplus \mathcal{R}_q^\epsilon(\mathcal{W}; \ell, m)$, where $\oplus$ denotes the Minkowski sum,

$$x \overset{\mathcal{W}}{\sim} \mathbf{I} \;\Rightarrow\; \Pr \big[\; \Pr[\; f(\mathbf{vec}(x)) \in \mathcal{R}_f^\epsilon(\mathcal{W}; \ell, m) \;] > 1 - \epsilon \;\big] > 1 - \mathsf{betacdf}_{1-\epsilon}(\ell, m + 1 - \ell).$$

This technique offers two key improvements over the naive approach:

- The reachset is no longer constrained to a hyper-rectangle, eliminating the first source of conservatism inherent in the naive method.

- The calibration dataset is now defined using prediction errors rather than the network's raw outputs. Since prediction errors are typically of much smaller magnitude than the outputs of model $f$, this significantly reduces the conservatism of conformal inference in high-dimensional spaces. This implies that a useful surrogate model must maintain an acceptable level of fidelity.

However, the small ReLU neural network proposed in Hashemi et al. (2025) introduces three main challenges for this methodology:

- The surrogate model may suffer from fidelity issues, yielding correct yet conservative probabilistic reachable sets.

- As the perturbation dimensionality grows, the ReLU surrogate's input dimension increases, creating scalability issues for training. In addition, deterministic reachability on the ReLU surrogate also scales poorly, often leading to out-of-memory errors in the NNV toolboxTran et al. (2020b). Finally, because NNV is restricted to polytopes, image perturbations must be constrained to $\ell_\infty$.

In the next section, we introduce the idea of **clipping-block** for constructing the surrogate model $g$. Unlike the previous method, our new approach requires no additional training—eliminating fidelity concerns—and imposes no restrictions on the perturbation of the input image.

### 3.1 GENERATING THE SURROGATE MODEL VIA CLIPPING-BLOCK

The clipping block is the core component of our surrogate model, providing both an accurate approximation of the original system and a deterministic reachable set. Specifically, we first generate a cloud of outputs from the training dataset and construct a convex hull enclosing these points. We formulate this convex hull as,

$$\mathbf{CH} = \{z \mid z = \sum_{j=1}^t \alpha_j \mathbf{vec}(y_j^{\text{train}}), \;\; \sum_{j=1}^t \alpha_j = 1, \quad \alpha_1, \cdots, \alpha_t > 0\}$$

The clipping block then applies a convex optimization procedure that projects every unseen output $\mathbf{vec}(y^{\text{unseen}})$ onto this hull, as $\mathbf{vec}(\hat{y}^{\text{unseen}}) = \sum_{j=1}^{t} \alpha_j^* \mathbf{vec}(y_j^{\text{train}})$, where,

$$\alpha_1^*, \cdots, \alpha_t^* = \underset{\alpha_1, \cdots, \alpha_t}{\mathbf{argmin}} \|\mathbf{vec}(y^{\text{unseen}}) - \sum_{j=1}^{t} \alpha_j \mathbf{vec}(y_j^{\text{train}})\|_l \quad \mathbf{s.t.} \ \sum_{j=1}^{t} \alpha_j = 1, \ \alpha_1, \cdots, \alpha_t > 0,$$

that for $l = 2$ is a quadratic programming and for $l = 1, \infty$ is a linear programming. Among the three options, we are most interested in $l = \infty$, since it minimizes the maximum error across the logit components, making it a suitable choice for reachability analysis. In addition, the $l = 2$ formulation is far more memory-intensive compared to the $l = \infty$ case, and LP formulation scales more gracefully. Each problem is linear, and batching helps amortize model builds, and parallelism is also effective — which is why we observe it running stably on our setup.

Thus, our surrogate model $g$ is composed of two main components. The first is the base model $f$, which takes the vectorized image $\mathbf{vec}(x^{\text{unseen}})$ and produces the corresponding logit vector $\mathbf{vec}(y^{\text{unseen}})$. The second is the clipping block, which takes $\mathbf{vec}(y^{\text{unseen}})$ as input and returns its projection onto the convex hull $\mathbf{CH}$, denoted as $\mathbf{vec}(\hat{y}^{\text{unseen}})$. In other words, if we denote the projection operation performed in the clipping block with $\mathbf{CLP}$, then,

$$g(\mathbf{vec}(x^{\text{unseen}})) = \mathbf{vec}(\hat{y}^{\text{unseen}}) = \mathbf{CLP}(f(\mathbf{vec}(x^{\text{unseen}})))$$

and obviously $\mathbf{CH}$ represents the deterministic reachable set of the surrogate model, $\mathbf{R}_g(\mathbf{I}) = \mathbf{CH}$. The availability of the surrogate reachset, regardless of the type or dimensionality of the input perturbation, removes the need to assume sparsity and eliminates any restrictions on the form of the image perturbation. Appendix C uses a toy example to make a comparison between our clipping block approach and the Naive technique on the reachability of deep neural networks.

To generate the convex hull, it is not necessary to include the entire training dataset. Instead, one can identify the extreme points Sartipizadeh & Vincent (2016) and construct the convex hull using only those points, which reduces the number of coefficients $\alpha$ without sacrificing accuracy. This approach makes the convex hull more efficient to be used as it reduces the number of coefficients $\alpha$. Existing algorithms for identifying extreme points include Quickhull Barber et al. (1996) and Clarkson's algorithm Clarkson (1988). Their computational complexities are $\mathcal{O}(t \log(t))$ for $n = 2, 3$ and $\mathcal{O}(t^{\lfloor n/2 \rfloor})$ for $n \geq 4$, which do not scale well to high-dimensional spaces. Another approach to this problem is to approximate the convex hull by allowing a controlled level of inaccuracy Sartipizadeh & Vincent (2016); Jia et al. (2022). However, as reported in the literature Casadio et al. (2025), this method has not shown promising performance even in high-dimensional settings such as large language models, whose dimensionality is still lower than that of semantic segmentation networks.

For this reason, we keep the original formulation and retain all points in the training dataset when constructing the convex hull. Importantly, this does not create scalability issues for our experiments, since conformal inference is highly data-efficient. For example, guarantees of order $\delta_1 = 0.999$ and $\delta_2 = 0.997$ can be verified with a calibration dataset of size $m = 8000$. Because we consider the size of the training dataset to be at most $50\%$ of the calibration dataset size, then $t = 4000$ training samples suffice for our experiments, requiring only $4000$ coefficients $\alpha$ in the convex hull formulation. Moreover, since the number of extreme points grows exponentially with the dimension of the logits, i.e., $\mathcal{O}((\log t)^{n-1})$ Dwyer (1988), it is highly likely that most training samples will serve as extreme points. This renders attempts to distinguish between extreme and interior points largely ineffective.

Similar to Hashemi et al. (2025), which encounters scalability issues when training a surrogate model in high-dimensional output spaces, our approach also suffers from this limitation. Specifically, projecting $\mathbf{vec}(y^{\text{unseen}})$ onto $\mathbf{CH}$ using linear programming does not scale efficiently with output dimensionality. To address this, the authors of Hashemi et al. (2025) introduce a scalable variant of Principal Component Analysis (PCA), known as the deflation algorithm, prior to training. We also found this strategy beneficial and adopt PCA before implementing the clipping block. Consequently, we conclude that the prediction error in our method arises primarily from the combined effects of PCA and the clipping block. The following section details this incorporation, which enables our technique to extend to robustness analysis of SSNs.

---

**Algorithm 1:** Learning Based Principal Component Analysis through Deflation Algorithm

---

**Initialize** $z_1$, $z_2$, ..., $z_t \leftarrow \mathbf{vec}(y_1^{\text{train}})$, $\mathbf{vec}(y_2^{\text{train}})$, ..., $\mathbf{vec}(y_t^{\text{train}})$, $A \leftarrow [\;]$

**for** index $= 0, 1, \ldots N - 1$ **do**

    // Train the principal direction using stochastic gradient ascent

    $\vec{a}_{\text{index}} \leftarrow \arg\max_{\vec{a}} \left[ \mathcal{J} := \frac{1}{t} \sum_{j=1}^{t} \vec{a}^{\top} z_j z_j^{\top} \vec{a} \right]$,     **s.t.**   $\|\vec{a}\|_2 = 1$

    $A.\mathbf{append}(\vec{a}_{\text{index}})$             // collect the principal direction in A

    // Update the dataset by component removal along principal direction.

    $z_1, z_2, \ldots, z_t \leftarrow z_1 - (\vec{a}_{\text{index}}^{\top} z_1)\vec{a}_{\text{index}}, \; z_2 - (\vec{a}_{\text{index}}^{\top} z_2)\vec{a}_{\text{index}}, \; \ldots, \; z_t - (\vec{a}_{\text{index}}^{\top} z_t)\vec{a}_{\text{index}}$

---

## 3.2 SURROGATE MODEL FOR SSNs VIA PRINCIPAL COMPONENT ANALYIS

In this section, we present our methodologies to mitigate the dimensionality challenges to provide the surrogate model, and present our technique for the robustness analysis of SSNs.

In our robustness analysis of SSNs, as indicated by equation equation 5, adversarial perturbations are confined to an $r$-dimensional subspace of the input space. Thus, we reformulate the surrogate as,

$$g'(\lambda) = g(\,\mathbf{vec}(\,x + \sum_{i=1}^{r} \lambda(i)x^{\text{noise}}\,)\,) = g(\mathbf{vec}(x^{\text{adv}})) \quad \text{and} \quad \lambda \in [\underline{\lambda}, \overline{\lambda}] \subset \mathbb{R}^r,$$

and also the deterministic reachset $\mathbf{R}_g(\mathbf{I})$ based on the vector $\lambda$ as $\mathbf{R}_{g'}([\underline{\lambda}, \overline{\lambda}]) = \mathbf{R}_g(\mathbf{I}) = \mathbf{CH}$.

**High-Dimensionality of Output Space**. To address this challenge, we firt reduce the dimensionality of the logits i.e., for a choice of $N \ll n$, and then generate the convex hull to be used in the clipping process. Let's sample training images $x_j^{\text{train}}$, for $j = 1, \ldots, t$, from the adversarial set $\mathbf{I}$, using sampling the coefficient vectors $\lambda_j^{\text{train}}$ form $[\underline{\lambda}, \overline{\lambda}]$. Then the corresponding logits $\mathbf{vec}(y_j^{\text{train}})$ form a point cloud in $\mathbb{R}^n$. This stage aims to train the top $N$ principal directions of cloud, which are stored as columns of the matrix $A = [\vec{a}_0, \vec{a}_1, \ldots, \vec{a}_{N-1}] \in \mathbb{R}^{n \times N}$.

We employ this matrix to compress the high-dimensional logits $y_j^{\text{train}}$ into a reduced representation $v_j \in \mathbb{R}^N$, defined as $v_j = A^{\top}\mathbf{vec}(y_j^{\text{train}})$. While Principal Component Analysis (PCA) is the standard tool for dimensionality reduction, its direct application does not scale effectively in very high-dimensional settings. To overcome this limitation, deflation-based methods have been developed (Algorithm 1). These algorithms extract principal components one at a time, ordered by significance (Mackey, 2008). After each dominant component is obtained, the dataset is projected onto its orthogonal complement, thereby eliminating its influence before proceeding to the next iteration.

In this paper, we employ a learning-based deflation algorithm, outlined in Algorithm 1. In this approach, the components of the principal direction are treated as trainable parameters, and the objective is to maximize the variance of the dataset along this direction. At each iteration, the optimization problem for learning the principal direction admits one global maximum, one global minimum, and $n - 2$ saddle points, where $n$ denotes the dimension of the logits $\mathbf{vec}(y_j^{\text{train}})$ Mackey (2008). Consequently, despite operating in a high-dimensional space, convergence to the global maximum is guaranteed.

Next we generate the convex hull over the cloud of point of the low dimensional representatives of logit $v_j, j = 1, \ldots, t$ and then apply the clipping block. Thus, for any unseen $\lambda$ we define,

$$g'(\lambda^{\text{unseen}}) = g(x^{\text{adv}}) = A\,\mathbf{CLP}(A^{\top}f(\mathbf{vec}(\,x + \sum_{i=1}^{r} \lambda^{\text{unseen}}(i)x^{\text{noise}}\,)))$$

In conclusion we present the surrogate model for $f$ as $g(x^{\text{adv}})$ which is a scalable choice for SSNs.

In addition, we leverage conformal inference to generate a hyper-rectangle that bounds the error $q(\mathbf{vec}(x)) = f(\mathbf{vec}(x)) - g(\mathbf{vec}(x))$ between the models $f(\mathbf{vec}(x))$ and $g(\mathbf{vec}(x))$ with provable guarantees. This hyper-rectangle is then used to inflate the convex hull and obtain the probabilistic

**Algorithm 2:** Detection of the pixel status

**Input:** $\mathbf{I}$, $x$, $f$, $\mathcal{W}$, $\epsilon$, Hyper-parameters$(\ell, m)$
**Output:** The status of pixels

// Project the reachset
$[\underline{y}, \overline{y}] \leftarrow \mathcal{R}_f^\epsilon(\mathcal{W}; \ell, m)$

**foreach** $(i, j) \in \{1{:}h\} \times \{1{:}w\}$ **do**
    $l^* \leftarrow \arg\max_l \underline{y}(i, j, l)$
    **if** $\underline{y}(i, j, l^*) \leq \max_{l \neq l^*} \overline{y}(i, j, l)$ **then**
        Label pixel $(i, j)$ as unknown
    **else if** $l^* = \mathsf{SSN}(x)(i, j)$ **then**
        Label pixel $(i, j)$ as robust
    **else**
        Label pixel $(i, j)$ as nonrobust

**return** *Pixel-wise labeling*

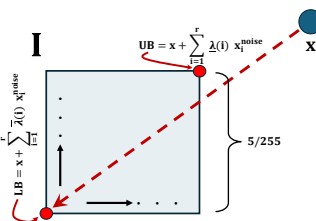

Figure 1: Illustrates the perturbation set $\mathbf{I}$ corresponding to an $r$-dimensional darkening adversary on image $x$. This set is made by independent perturbations on all $nc$ channels in $r'$ different pixels ($r = nc \times r'$), each with R,G, and B intensities greater than $150/255$. The lower bound $\mathbf{LB}$ represents the maximum darkening (channel intensities set to zero), while the upper bound $\mathbf{UB}$ corresponds to minimum darkening.

reachset. Here, we formulate this inflating hyper-rectangle as a convex hull:

$$\mathbf{HR} = \mathcal{R}_q^\epsilon(\mathcal{W};\ \ell, m) = \{z \mid z = c + \sum_{i=1}^n \beta_i \sigma_i e_i, \qquad \sum_{i=1}^n \beta_i = 1, \quad \beta_1, \cdots, \beta_n > 0\}$$

where $e_i \in \mathbb{R}^n$ is the $i$-th Cartesian unit vector in Euclidean space, $\sigma_i$ is the magnitude obtained from conformal inference (see Eq. equation 9) along direction $e_i$ and $c$ is the center of the hyper-rectangle.

Given this formulation, we compute the Minkowski sum between the deterministic reach set of the surrogate model $g$ and the inflating hyper-rectangle as:

$$\mathcal{R}_f^\epsilon(\mathcal{W}; \ell, m) = \mathcal{R}_g(\mathbf{I}) \oplus \mathcal{R}_q^\epsilon(\mathcal{W};\ \ell, m) = \mathbf{CH} \oplus \mathbf{HR} =$$

$$\{z \mid z = c + \sum_{j=1}^t \alpha_j A\, v_j^{\text{train}} + \sum_{i=1}^n \beta_i \sigma_i e_i,\ \ \alpha_1, \cdots, \alpha_t, \beta_1, \cdots, \beta_n \geq 0,\ \sum_{j=1}^t \alpha_j = 1,\ \sum_{i=1}^n \beta_i = 1\}$$

The projection of the probabilistic reach set $\mathcal{R}_f^\epsilon(\mathcal{W}; \ell, m)$ onto each class of a given pixel can be performed efficiently. To this end, we collect the training points $v_j^{\text{train}}, ; j = 1, \ldots, t$. Since the mapping from latent space to the original space is linear ($\mathbf{vec}(\hat{y}) = Av$), the vertices of the convex hull in latent space are preserved under this transformation. In other words, if $v^*$ is an extreme point of the convex hull in latent space, then $Av^*$ is an extreme point of the convex hull in the original space. Therefore, we collect the points $\mathbf{vec}(\hat{y}_j^{\text{train}}) = Av_j^{\text{train}}$ and compare them to determine the upper bound $\mathbf{ub}$ and lower bound $\mathbf{lb}$ of the surrogate model's predictions. Consequently, the clipping block in the surrogate model ensures that every unseen point $\mathbf{vec}(\hat{y}^{\text{unseen}})$ satisfies, $\mathbf{vec}(\hat{y}^{\text{unseen}}) \in [\mathbf{lb}, \mathbf{ub}]$. For a given pixel and class, let the corresponding logit value be the $k$-th component of $\mathbf{vec}(y^{\text{unseen}})$. Then the projection of the reachset on this component satisfies: $\mathbf{vec}(y^{\text{unseen}})(k) \in [c(k) + \mathbf{lb}(k) - \sigma_k\ ,\ c(k) + \mathbf{ub}(k) + \sigma_k]$.

We stress that setting $\mathcal{W}' = \mathcal{W}$ and $t = m/2$ markedly improves the surrogate model's fidelity in our new approach a setting we always follow in our experiments via clipping block technique.

## 4 EXPERIMENTS

In our numerical evaluation, we pose two different research questions and address each of them using numerical results. The main one is available in this section and the rest of them are in Appendix D. We utilized a Linux machine, with 48 GB of GPU memory, 512 GB of RAM, and 112 CPUs.

**RQ1: How does the technique scale with model size and dimensionality, number of perturbed pixels, and perturbation level?** To assess scalability, we test our verification method on high-dimensional datasets (e.g., Lung Segmentation (Jaeger et al., 2014; Candemir et al., 2014), OCTA-500 (Li et al., 2019), CamVid (Brostow et al., 2009)) using large pre-trained models. We evaluate

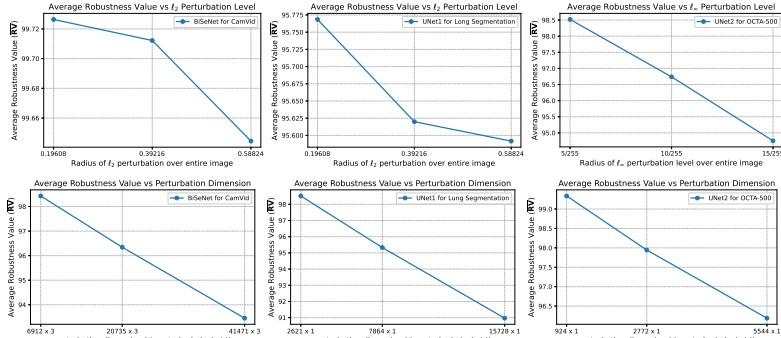

Figure 2: Shows $\overline{\mathbf{RV}}$ versus perturbation level and perturbation dimension. In the top row, we perturbed the entire image with $\ell_2$ perturbation for CamVid and Lung Segmentation and $\ell_\infty$ for OCTA-500. In the bottom row, we consider darkening adversary with $e = 5/255$ and we increase the dimension of perturbation. Robustness is averaged over 200 test images .

Table 1: Model details and probabilistic guarantees. Given hyperparameters $\ell$, $m$, and coverage $\delta_1 = 1 - \epsilon$, the confidence $\delta_2$ is computed for each experiment. The reported verification runtime is averaged over all experiments run on a model. One important point is, our verification runtime depends mainly on inference time, and hyperparameters $(\ell, m)$, and is only slightly sensitive to perturbation magnitude and dimension $(e, r)$. For per-experiment runtimes, see Figure 4 in Appendix D.

| Dataset name | Model name | Input dimension | Output dimension | Number of Parameters | $(m, m-\ell)$ | coverage $\delta_1$ | confidence $\delta_2$ | Average runtime (method) |
|---|---|---|---|---|---|---|---|---|
| Lung Segmentation | UNet1 | $512 \times 512 \times 1$ | $512 \times 512 \times 1$ | 14,779,841 | (8e3, 1) | 0.999 | 0.997 | 5.6 min (Surrogate) |
| OCTA-500 | UNet2 | $304 \times 304 \times 1$ | $304 \times 304 \times 1$ | 5,478,785 | (8e3, 1) | 0.999 | 0.997 | 9.8 min (Surrogate) |
| CamVid | BiSeNet | $720 \times 960 \times 3$ | $720 \times 960 \times 12$ | 12,511,084 | (8e3, 1) | 0.999 | 0.997 | 18.3 min (Surrogate) |
| Cityscapes (Appendix D) | HRNetV2 | $1024 \times 2048 \times 3$ | $256 \times 512 \times 19$ | 65,859,379 | (921, 1) | 0.99 | 0.997 | 3.5 min (Naive) |

performance on a subset of 200 test images across varying perturbation dimensions and magnitudes, with results shown in Figure 2. Model details and probabilistic guarantees are summarized in Table 1, where we show the following $\langle \epsilon, \ell, m \rangle$ guarantee[2]:

$$\Pr\left[\Pr\left[\mathsf{P}\right] \geq \delta_1\right] \geq \delta_2, \quad \text{where } \mathsf{P} := \text{''} \left\{ \begin{array}{c} \text{Given an image } x, \text{ the perturbation magnitude and dimension } e, r, \\ \text{the computed robustness value } \mathbf{RV} \text{ from our technique is valid.} \end{array} \right\} \text{''}$$

In these experiments, we studied an $r$-dimensional darkening adversary (see Figure 1), where $r'$ pixels in an image $x$ with intensity above $150/255$ in all channels are randomly selected for perturbation ($r = nc \times r'$). Each direction $x_i^{\text{noise}}$ corresponds to darkening a channel of one such pixels, and $x^{\text{adv}}$ is parameterized by an $r$-dimensional coefficient vector $\lambda \in [\underline{\lambda}, \overline{\lambda}]$. Here, $\overline{\lambda}$ induces full darkening (intensity zero), while $\underline{\lambda}$ applies partial darkening. This bound defines the perturbation space $x^{\text{adv}} \in \mathbf{I}$. We also present a demo for the status of all pixels in the segmentation mask in Figures 5,6 and 7.

To assess the conservatism, we examine the projection bounds $[\underline{y}, \overline{y}]$ introduced in Algorithm 2. Specifically, we sample $10^6$ adversarial examples from $x^{\text{adv}} \overset{\mathcal{W}}{\sim} \mathbf{I}$ and use them to report:

• Empirical miscoverage $\hat{\epsilon}$, that is the percentage of events, where $f(\mathbf{vec}(x^{\text{adv}})) \notin [\underline{y}, \overline{y}]$.

• Empirical bound $[\underline{\hat{y}}, \overline{\hat{y}}]$, via component-wise minima/maxima of $f(\mathbf{vec}(x^{\text{adv}}))$ across samples.

The degree of conservatism can be assessed by comparing $[\underline{\hat{y}}, \overline{\hat{y}}]$ with $[\underline{y}, \overline{y}]$. To measure this, we compute $\mathsf{bound\_ratio} = \sum_{k=1}^{h \times w \times L} (\overline{\hat{y}}(k) - \underline{\hat{y}}(k)) / \sum_{k=1}^{h \times w \times L} (\overline{y}(k) - \underline{y}(k))$ and due to the high cost of $10^6$ inferences on the models,we only perform the conservatism analysis on one specific case with BiSeNet from Figure 2, Table 3 in Appendix D details the numerical results.

## 5 CONCLUSION

Our verification approach is designed to be scalable and efficient. Although it does not provide deterministic guarantees, it offers strong probabilistic assurances in regions where deterministic methods are computationally intractable. This effectiveness is further demonstrated through the numerical experiments included in the paper.

---

[2]The surrogate-based technique in Hashemi et al. (2025) could not handle our perturbation dimensions.

## 6 REPRODUCIBILITY

We included anonymous toolbox as the supplementary material to be used for reproducibility.

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

# A A COMPARISON BETWEEN APPLICATION OF CI AND RANDOMIZED SMOOTHING ON ROBUSTNESS ANALYSIS OF SSNs.

In this section, we present a comparison with the works of Fiacchini & Alamo (2021); Anani et al. (2024) on the Cityscapes dataset Cordts et al. (2016). Since the formulation of verification guarantees differs slightly across these approaches, we first provide a general overview of the techniques proposed in Fiacchini & Alamo (2021); Anani et al. (2024), followed by a brief description of CI-based technique. This will clarify how the comparison can be meaningfully established.

**Overview 1**. In Fischer et al. (2021), the authors introduced a probabilistic method to verify Semantic Segmentation Neural Networks. Inspired by Cohen et al. (2019), they introduced randomness via a Gaussian noise $\nu \sim \mathcal{N}(0, \sigma^2) \in \mathbb{R}^{h \times w \times nc}$, applied to the input, and constructed a smoothed version of the segmentation model, denoted $\overline{\mathsf{SSN}}(x)(i,j)$, for each mask pixel $(i,j)$:

$$\overline{\mathsf{SSN}}(x)(i,j) = c_A(i,j) = \arg\max_{c \in \mathbf{L}} \Pr_{\nu \sim \mathcal{N}(0,\sigma^2)} [\mathsf{SSN}(x+\nu)(i,j) = c]$$

They then established that, with confidence level $\delta_2$, the smoothed prediction $\overline{\mathsf{SSN}}(x)(i,j)$ is robust within an $\ell_2$ ball $\mathbf{B}_{\bar{r}_{i,j}}(x)$ of radius $\bar{r}_{i,j} = \sigma \Phi^{-1}(\underline{p_A}(i,j))$, where $\underline{p_A}(i,j)$ is a lower bound on the class probability: $p_A(i,j) = \Pr_{\nu \sim \mathcal{N}(0,\sigma^2)} [\mathsf{SSN}(x+\nu)(i,j) = c_A(i,j)]$. Here, $\Phi^{-1}(.)$ is the inverse CDF of normal distribution. The corresponding probabilistic guarantee becomes:

$$\forall x' \in \mathbf{B}_{\bar{r}_{i,j}}(x) : \Pr\left[\overline{\mathsf{SSN}}(x')(i,j) = c_A(i,j)\right] \geq \delta_2$$

However, certifying all mask pixels simultaneously is challenging. To enable global guarantees, a more conservative definition of smoothing is adopted: a fixed threshold $\tau \in [0.5, 1]$ is used for all mask locations $(i,j)$ which implies:

$$\overline{\mathsf{SSN}}(x)(i,j) = c_A(i,j), \quad \text{where} \quad \Pr_{\nu \sim \mathcal{N}(0,\sigma^2)} [\mathsf{SSN}(x+\nu)(i,j) = c_A(i,j)] \geq \tau$$

This leads to a unified radius $r = \sigma \Phi^{-1}(\tau)$ for the input space. While this simplifies the formulation, it introduces the concept of **abstained** or uncertifiable pixels—those that cannot meet the desired confidence threshold. To mitigate the issue of multiple comparisons (i.e., union bounds), the authors propose statistical corrections such as the Bonferroni and Holm–Bonferroni methods CE (1936); Holm (1979). Assuming the set of certifiable pixels is defined as $\mathbf{CERT} = \{(i,j) \mid (i,j) \text{ is certifiable}\}$, they propose the final guarantee in the following form:

$$\forall x' \in \mathbf{B}_r(x) : \Pr\Big[\bigwedge_{(i,j) \in \mathbf{CERT}} \mathsf{P}(i,j)\Big] \geq \delta_2, \quad \mathsf{P}(i,j) := \left\{"\overline{\mathsf{SSN}}(x')(i,j) = c_A(i,j)"\right\}$$

This method is referred to as SEGCERTIFY. A key limitation of this technique is the presence of abstained pixels. The authors in Anani et al. (2024) addressed this by proposing an adaptive approach, ADAPTIVECERTIFY, which reduces the number of uncertifiable pixels. However, the guarantees proposed in Fiacchini & Alamo (2021); Anani et al. (2024) apply only to the smoothed model, not the base model. To express this in terms of the base model, we define the following problem:

*Problem 1*. For a given image $x$, noise $\nu \sim \mathcal{N}(0, \sigma^2)$, and a threshold $\tau \in [0.5, 1]$, define $c_A(i,j) \in \mathbf{L}$ such that: $\Pr_{\nu \sim \mathcal{N}(0,\sigma^2)} [\mathsf{SSN}(x+\nu)(i,j) = c_A(i,j)] \geq \tau$. Then for any $x' \in \mathbf{B}_r(x)$ with $r = \sigma \Phi^{-1}(\tau)$ and confidence level $\delta_2$, we want to show the following guarantee, where **CERT** will be also determined through the verification process:

$$\Pr\Big[\bigwedge_{(i,j) \in \mathbf{CERT}} \Pr_{\nu \sim \mathcal{N}(0,\sigma^2)} [\mathsf{SSN}(x'+\nu)(i,j) = c_A(i,j)] \geq \tau\Big] \geq \delta_2$$

**Overview 2**. In CI approach, for a given image $x$ and input set $\mathbf{B}_r(x)$, we perform a probabilistic reachability analysis with a $\langle \epsilon, \ell, m \rangle$ guarantee. Due to the presence of conservatism in the

Table 2: Percentage of uncertifiable pixels under different $(\sigma, \tau, r)$ settings for Fiacchini & Alamo (2021), Anani et al. (2024), and the Naive approach. The verification runtime is 210 seconds for all experiments.

| $\sigma$ | $\tau$ | $r$ | Fiacchini & Alamo (2021) (%) | Anani et al. (2024) (%) | Naive approach (%) |
|---|---|---|---|---|---|
| 0.25 | 0.75 | 0.1686 | 7 | 5 | 0.0642 |
| 0.33 | 0.75 | 0.2226 | 14 | 10 | 0.0676 |
| 0.50 | 0.75 | 0.3372 | 26 | 15 | 0.0705 |
| 0.25 | 0.95 | 0.4112 | 12 | 9 | 0.0727 |
| 0.33 | 0.95 | 0.5428 | 22 | 18 | 0.0732 |
| 0.50 | 0.95 | 0.8224 | 39 | 28 | 0.0758 |

reachability technique, some mask pixels are marked as **robust** (i.e., certifiable), while others cover multiple classes and are considered **unknown** or uncertifiable. Let $\textbf{CONFORMAL\_CERT} = \{(i,j) \mid (i,j) \text{ is certifiable}\}$. Then the verification objective is:

*Problem 2.* Given an image $x$, input set $\mathbf{B}_r(x)$, and sampling distribution $x' \overset{\mathcal{W}}{\sim} \mathbf{B}_r(x)$, for coverage level $\delta_1 = 1 - \epsilon \in [0,1]$, hyper-parameters $\ell, m$ and confidence level $\delta_2 = 1 - \mathsf{betacdf}_{\delta_1}(\ell, m+1-\ell)$, we aim to show the following $\langle \epsilon, \ell, m \rangle$ guarantee, where **CONFORMAL\_CERT** will be also determined through the verification process:

$$\Pr \Big[ \Pr \big[ \bigwedge_{(i,j) \in \textbf{CONFORMAL\_CERT}} \mathsf{SSN}(x')(i,j) = \mathsf{SSN}(x)(i,j) \big] \geq \delta_1 \Big] \geq \delta_2$$

**Comparison**. The unknown pixels in the CI-based technique are conceptually equivalent to the abstained pixels in Fiacchini & Alamo (2021); Anani et al. (2024). Thus, given the same input set $\mathbf{B}_r(x)$ where $r$ is provided by Fiacchini & Alamo (2021), the comparison focuses to show which technique results in fewer uncertifiable mask pixels. To this end, we replicate the setup of Table 1 from Anani et al. (2024), using the same HrNetV2 model Wang et al. (2020)trained on the Cityscapes dataset with the HrNetV2-W48 backbone. In this case study, the results of the Naive approach were approximately equivalent to those of the surrogate-based approach. Therefore, we use the Naive technique for comparison, as it is more efficient than the surrogate-based method. We compare the number of uncertifiable pixels produced by the naive approach against the abstained pixels reported in their results. Our findings are summarized in Table 2, which shows the average over 200 test images. We also report the verification runtime for completeness. It is important to note that although the number of uncertifiable pixels in CI-based method is significantly smaller than in prior approaches, the CI-based formulation of guarantees also differs. In particular, we assume a prior distribution for the uncertainty, where $\mathcal{W}$ is considered to be uniform. This assumption, however, can be readily extended to worst-case distribution analysis by replacing conformal inference with robust conformal inference Cauchois et al. (2024), a technique that strengthens guarantees at a modest cost. We leave this extension to future work.

## B  NAIVE REACHABILITY TECHNIQUE VIA CONFORMAL INFERENCE.

The authors in Hashemi et al. (2025) first construct a probabilistic reachable set for neural networks solely using the conformal inference, yielding a hyper-rectangular reachset. To achieve this, they firstly generate a calibration dataset $\mathbf{M}$ of size $m$ along with a training dataset $\mathbf{T}$ of size $t$.

To construct the training dataset, $t$ inputs $x_j^{\text{train}}, ; j = 1, 2, \ldots, t$ are sampled from $\mathbf{I}$ according to any chosen distribution, denoted $x \overset{\mathcal{W}'}{\sim} \mathbf{I}$. Their corresponding outputs are then computed as $\mathbf{vec}(y_j^{\text{train}}) = f(\mathbf{vec}(x_j^{\text{train}}))$. The resulting collection forms the training dataset, $\mathbf{T} = \{(x_1^{\text{train}}, y_1^{\text{train}}), (x_2^{\text{train}}, y_2^{\text{train}}), \ldots, (x_t^{\text{train}}, y_t^{\text{train}})\}$.

To construct the calibration dataset, $m$ inputs $x_i^{\text{calib}}, ; i = 1, 2, \ldots, m$ are sampled from $\mathbf{I}$ according to the distribution $x \overset{\mathcal{W}}{\sim} \mathbf{I}$. Their corresponding outputs are computed as $\mathbf{vec}(y_i^{\text{calib}}) = f(\mathbf{vec}(x_i^{\text{calib}}))$. This yields the input/output dataset $\mathbf{IO} = \{(x_1^{\text{calib}}, y_1^{\text{calib}}), (x_2^{\text{calib}}, y_2^{\text{calib}}), \ldots, (x_m^{\text{calib}}, y_m^{\text{calib}})\}$. This dataset is then used to compute a suitable nonconformity score $R \in \mathbb{R}_{\geq 0}$. Motivated by recent

applications of conformal inference in time-series Cleaveland et al. (2024); Hashemi et al. (2024), the authors design the nonconformity scores to produce a hyper-rectangular set. For an output $\mathbf{vec}(y) = [\mathbf{vec}(y)(1), \ldots, \mathbf{vec}(y)(n)] \in \mathbb{R}^n$, Hashemi et al. (2025) selects the calibration scores as,

$$R_i^{\text{calib}} = \max\left(\frac{|\mathbf{vec}(y_i^{\text{calib}})(1) - c(1)|}{\tau_1}, \ldots, \frac{|\mathbf{vec}(y_i^{\text{calib}})(n) - c(n)|}{\tau_n}\right), \quad i = 1, 2, \ldots, m, \quad (6)$$

where $c = \sum_{j=1}^t \mathbf{vec}(y_j^{\text{train}})$ is the average of the outputs in $\mathbf{T}$. For each coordinate $k = 1, 2, \ldots, n$, the normalization factor $\tau_k$ rescales the random variables $|\mathbf{vec}(y) - c|$ and is computed from the training dataset as

$$\tau_k := \max\left(\tau^*, \ \max\left(|\mathbf{vec}(y_1^{\text{train}})(k) - c(k)|, \ldots, |\mathbf{vec}(y_t^{\text{train}})(k) - c(k)|\right)\right), \quad (7)$$

with $\tau^* = 10^{-5}\left(\sum_{k=1}^n \sum_{j=1}^t |\mathbf{vec}(y_j^{\text{train}})(k) - c(k)|\right)/nt$ introduced to prevent division by zero. Finally, the calibration dataset is given by $\mathbf{M} = \{R_1^{\text{calib}}, R_2^{\text{calib}}, \ldots, R_m^{\text{calib}}\}$.

Following the principles of conformal inference, the nonconformity scores in $\mathbf{M}$ are first sorted in ascending order. Without loss of generality, assume $R_1^{\text{calib}} < R_2^{\text{calib}} < \cdots < R_m^{\text{calib}}$. Now, consider a new sample $x^{\text{unseen}}$ drawn from the distribution $x \overset{\mathcal{W}}{\sim} \mathbf{I}$. Given the neural network architecture, the corresponding output is $\mathbf{vec}(y^{\text{unseen}}) = f(\mathbf{vec}(x^{\text{unseen}}))$, which follows some distribution denoted by $y^{\text{unseen}} \sim \mathcal{Y}$. Through the proposed mapping for nonconformity scores, the associated random variable is

$$R^{\text{unseen}} = \max\left(\frac{|\mathbf{vec}(y^{\text{unseen}})(1) - c(1)|}{\tau_1}, \ldots, \frac{|\mathbf{vec}(y^{\text{unseen}})(n) - c(n)|}{\tau_n}\right). \quad (8)$$

This induces another distribution, written $R \sim \mathcal{D}$. Both $R^{\text{unseen}}$ and the nonconformity scores $R_1^{\text{calib}}, R_2^{\text{calib}}, \ldots, R_m^{\text{calib}}$ are all i.i.d. samples from $\mathcal{D}$, which ensures the applicability of conformal inference. Consequently, for the new draw $R^{\text{unseen}}$, given a desired rank $\ell \leq m$ and miscoverage level $\epsilon$, Hashemi et al. (2025) chooses the following $\langle \epsilon, \ell, m \rangle$ guarantee:

$$\Pr\left[\Pr\left(R^{\text{unseen}} \leq R_\ell^{\text{calib}}\right) > 1 - \epsilon\right] > 1 - \mathsf{betacdf}_{1-\epsilon}(\ell, m + 1 - \ell).$$

From Equation equation 8, the condition $R^{\text{unseen}} \leq R_\ell^{\text{calib}}$ can be expressed as the logical statement $P_1(\ell, m) := [R^{\text{unseen}} \leq R_\ell^{\text{calib}}]$. This condition is equivalent to requiring that every coordinate of $\mathbf{vec}(y^{\text{unseen}})$ lies within a bounded interval around $c(k)$, namely

$$P_2(\ell, m) := \bigwedge_{k=1}^n \left[c(k) - \sigma_k \leq \mathbf{vec}(y^{\text{unseen}})(k) \leq c(k) + \sigma_k\right], \quad (9)$$

where $\sigma_k = \tau_k R_\ell^{\text{calib}}$. This describes a hyper-rectangular region that serves as the reachset for unseen outputs of the neural network sampled from $x \overset{\mathcal{W}}{\sim} \mathbf{I}$. Since $P_1(\ell, m)$ carries the $\langle \epsilon, \ell, m \rangle$ guarantee, the equivalent hyper-rectangular reachset inherits the same coverage guarantee for unseen outputs.

**Advantages and Disadvantages for Application of CI on SSN**  Reasoning directly about the output distribution $y \sim \mathcal{Y}$ resulting from inputs $x \overset{\mathcal{W}}{\sim} \mathbf{I}$ is generally infeasible due to the strong nonlinearity of neural networks. Consequently, providing probabilistic coverage guarantees over the network's output space becomes highly challenging. Conformal inference (CI) offers a key advantage in this setting: its guarantee is distribution free, meaning they remain valid for any distribution family that adequately represents the calibration dataset. This property makes CI particularly suitable for neural network reachability analysis, where output distributions are often complex.

Despite this, applying CI to high-dimensional outputs introduces significant difficulties. The scalar nonconformity scores $R$ in the calibration dataset are defined based on the network outputs $y \sim \mathcal{Y}$, but in high dimensions, the family of distributions $y \sim \mathcal{Y}$ that can generate the calibration distribution $R \sim \mathcal{D}$ grows dramatically in size. As a result, CI is robust to all the members of this family and thus tends to produce overly conservative guarantees, which limits its practical applicability to networks like semantic segmentation networks (SSNs) that generate extremely high-dimensional outputs. To overcome this limitation, Hashemi et al. (2025) propose a new learning-based reachability analysis method for SSNs scalable for high dimensional settings.

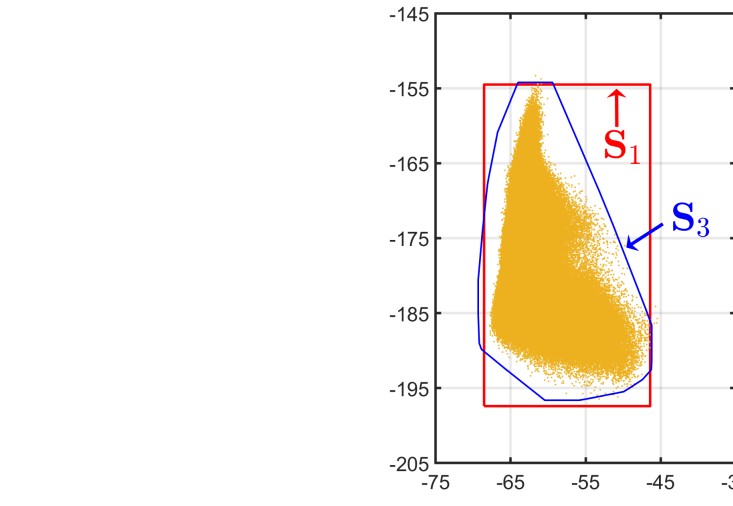

Figure 3: The figure compares two reachable sets with the same $\langle \epsilon, \ell, m \rangle$ guarantees. $S_1$ is obtained using the naïve approach, while $S_3$ is produced by the surrogate-based technique, illustrating the accuracy of the latter.

## C  TOY EXAMPLE

In this section we provide a comparison against the naïve approach from Hashemi et al. (2025). Specifically, we consider the toy example proposed in Hashemi et al. (2025) provided below.

**Example C.1** *Consider a feedforward* ReLU *neural network with* 60 *hidden-layers of width* 100*, an input layer of dimension* 784 *and an output layer of dimension* 2*. The set of inputs is* $\mathbf{I} := [0, 1]^{784}$ *and we generate both calibration and train datasets by sampling* $\mathbf{I}$ *uniformly. e.g.,* $\mathcal{W}, \mathcal{W}'$ *are both uniform distributions. We plan to generate reachable sets for this model that satisfies the following* $\langle \epsilon, \ell, m \rangle$ *guarantee with our technique and naive approach.*

$$x \overset{\mathcal{W}}{\sim} \mathbf{I} \;\Rightarrow\; \Pr[\; \Pr[\; f(x) \in \mathbf{S}_1 \;] > 0.9999 \;] > 0.9999995. \tag{10}$$

*To solve this problem Hashemi et al. (2025) generates a calibration dataset of size* $m = 200,000$ *and a train dataset of size* $t = 10,000$*, they also consider the rank* $\ell = 199,998$ *and target the miscoverage level of* $\epsilon = 0.0001$*. In this case they compute the reachable set,* $\mathbf{S}_1$ *presented in Figure 3 in orange.*

*On the other hand, we collect a calibration dataset of size* $m = 200,000$ *and compare the resulting bounds. For this comparison, we employ a clipping block that projects with* $l = \infty$*. We constructed the convex hull using a training dataset of size* $t = 4000$*. In addition, we sampled a separate dataset of size* $t' = 10000$ *to compute the normalization factors (see Eq. equation 7) prior to performing conformal inference for obtaining the inflating hyper-rectangle. The resulting reachable set* $\mathbf{S}_3$*, together with the outcome of the naïve approach, in the presence of* $10^6$ *new simulations,* $f(x), x \overset{\mathcal{W}}{\sim} \mathbf{I}$*, are shown in Figure 3.*

*Our calculations show that the proportion of points lying outside of* $\mathbf{S}_1$ *and* $\mathbf{S}_3$ *are 0.000013 and 0.000012, respectively, which aligns well with the proposed* $\langle \epsilon, \ell, m \rangle$ *guarantees.*

## D  ADDITIONAL RESEARCH QUESTIONS

**RQ2: Do State-of-the-Art Deterministic Verification Techniques Scale to the Level Achieved by Our Method on Complex and High-Dimensional SSN Models?**   We applied techniques such as $\alpha$-$\beta$-CROWN (Zhou et al., 2024) and NNV (Tran et al., 2021) to our segmentation experiments using the UNet1, UNet2, and BiSeNet models in RQ1. However, due to the size of these models and

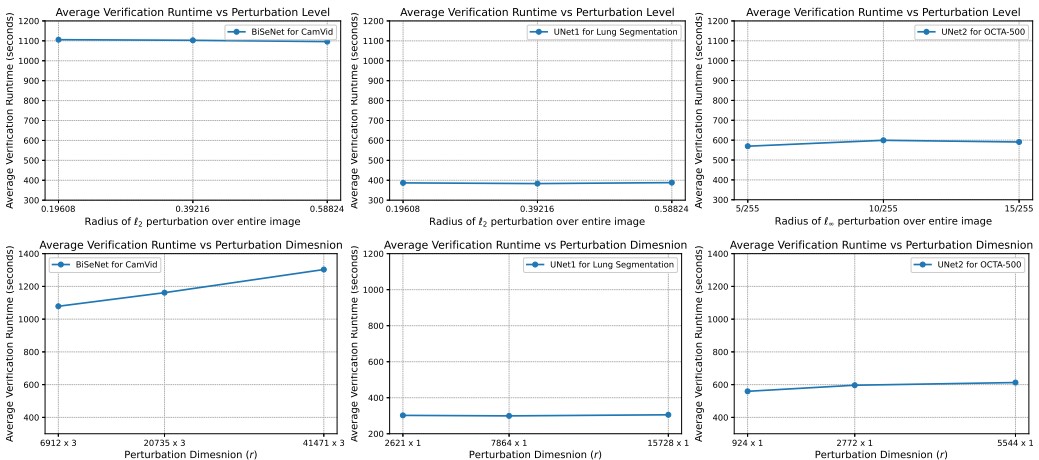

Figure 4: Shows the average run time versus perturbation level (top row) and perturbation dimension $r$ (bottom row). In the top row, the image is entirely perturbed within an $\ell_2$ ball for CamVid and Lung Segmentation and $\ell_\infty$ ball for OCTA-500. In the bottom row, we have darkening adversary where $e$ is fixed at $5/255$ across all experiments. The runtimes are averaged over 200 test images from datasets.

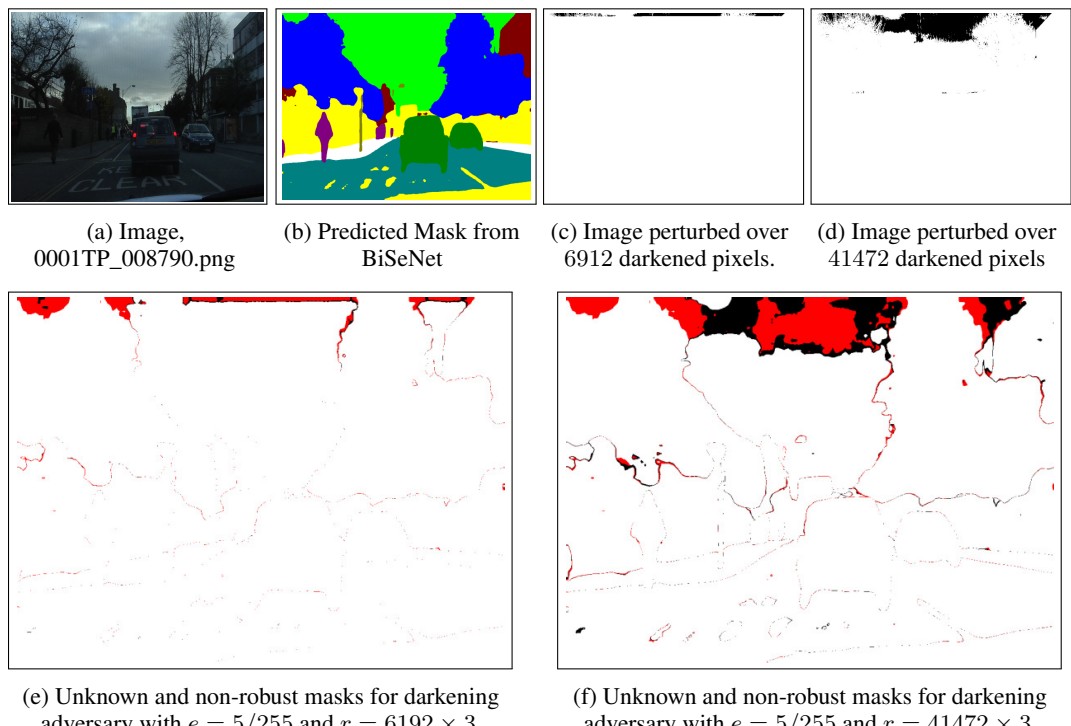

(a) Image, 0001TP_008790.png

(b) Predicted Mask from BiSeNet

(c) Image perturbed over 6912 darkened pixels.

(d) Image perturbed over 41472 darkened pixels

(e) Unknown and non-robust masks for darkening adversary with $e = 5/255$ and $r = 6192 \times 3$.

(f) Unknown and non-robust masks for darkening adversary with $e = 5/255$ and $r = 41472 \times 3$.

Figure 5: Visualization for verification on BiSeNet for a test image from the CamVid dataset. (a) The test image used for verification. (b) The segmentation mask predicted by BiSeNet for this image. (c,d) The pixels selected for perturbation (in black) on the test image (We sampled 6192 (1% of) and 41472 (6% of) pixels where the R, G, and B intensities were all above $150/255$, forming a perturbation set $\mathbf{I} \subset \mathbb{R}^{6192 \times 3}$ and $\mathbf{I} \subset \mathbb{R}^{41472 \times 3}$ respectively). (e,f) Display the robust (white), non-robust (red), and unknown (black) pixels for the perturbation set $\mathbf{I}$ as described in Figure 1, with perturbation magnitudes $e = 5/255$ and perturbation dimension $r = 6192 \times 3$ and $r = 41472 \times 3$, respectively.

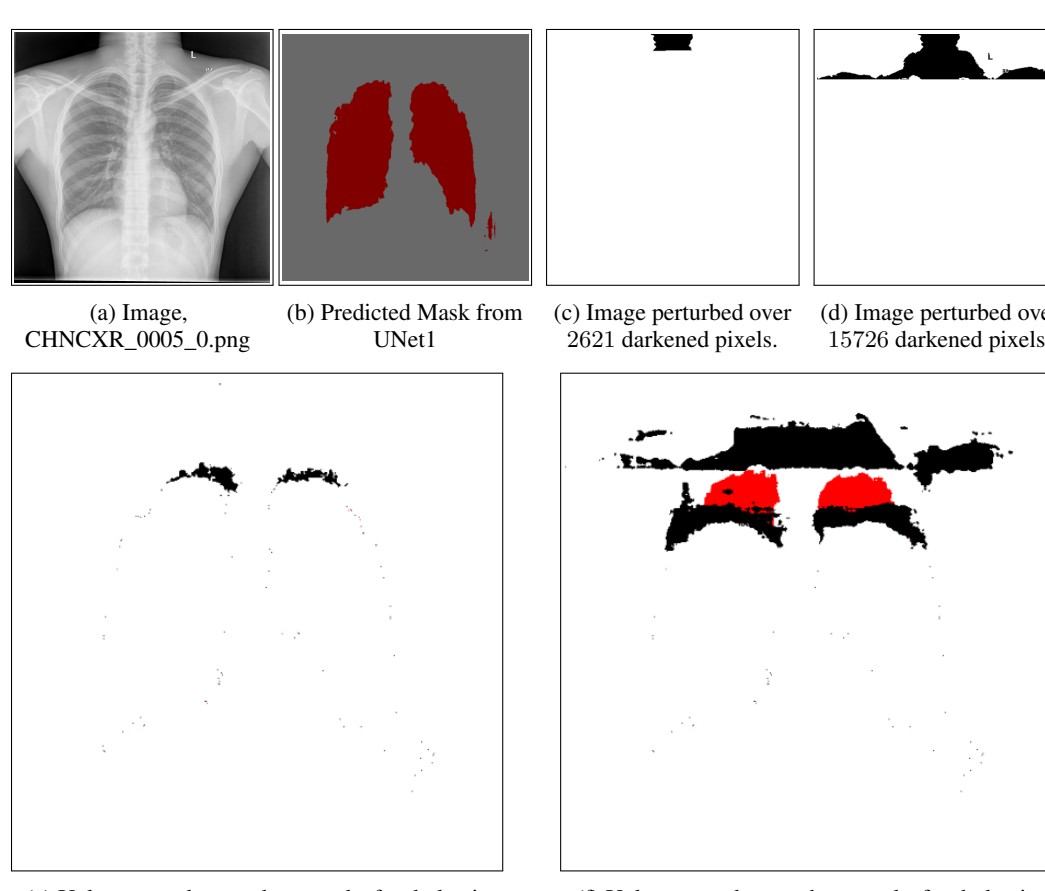

(a) Image, CHNCXR_0005_0.png

(b) Predicted Mask from UNet1

(c) Image perturbed over 2621 darkened pixels.

(d) Image perturbed over 15726 darkened pixels.

(e) Unknown and non-robust masks for darkening adversary with $e = 5/255$ and $r = 2621 \times 1$.

(f) Unknown and non-robust masks for darkening adversary with $e = 5/255$ and $r = 15726 \times 1$.

Figure 6: Visualization for verification on UNet1 for a test image from the Lung Segmentation dataset. (a) The test image used for verification. (b) The segmentation mask predicted by UNet1 for this image. (c,d) The pixels selected for perturbation (in black) on the test image (We sampled 2621(1% of) and 15726 (6% of) pixels where the Gray intensities were above $150/255$, forming a perturbation set $\mathbf{I} \subset \mathbb{R}^{2621 \times 1}$ and $\mathbf{I} \subset \mathbb{R}^{15726 \times 1}$ respectively). (e,f) Display the robust (white), non-robust (red), and unknown (black) pixels for the perturbation set $\mathbf{I}$ as described in Figure 1, with perturbation magnitudes $e = 5/255$ and perturbation dimnesion $r = 2621 \times 1$ and $r = 15726 \times 1$, respectively.

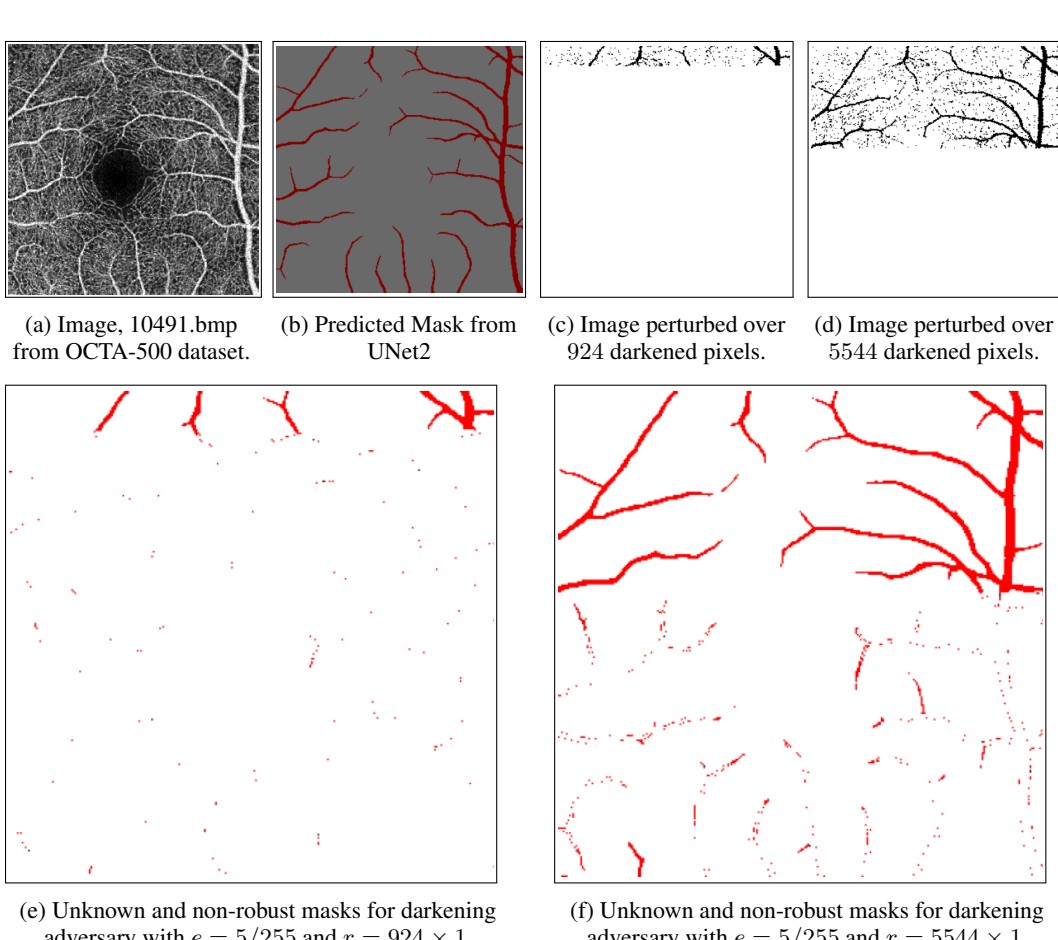

(a) Image, 10491.bmp from OCTA-500 dataset.

(b) Predicted Mask from UNet2

(c) Image perturbed over 924 darkened pixels.

(d) Image perturbed over 5544 darkened pixels.

(e) Unknown and non-robust masks for darkening adversary with $e = 5/255$ and $r = 924 \times 1$.

(f) Unknown and non-robust masks for darkening adversary with $e = 5/255$ and $r = 5544 \times 1$.

Figure 7: Visualization for verification on UNet2 for a test image from the OCTA-500 dataset. (a) The test image used for verification. (b) The segmentation mask predicted by UNet2 for this image. (c,d) The pixels selected for perturbation (in black) on the test image (We sampled $924(1\%$ of) and $5544$ (6% of) pixels where the Gray intensities were above $150/255$, forming a perturbation set $\mathbf{I} \subset \mathbb{R}^{924 \times 1}$ and $\mathbf{I} \subset \mathbb{R}^{5544 \times 1}$ respectively). (e,f) Display the robust (white), non-robust (red), and unknown (black) pixels for the perturbation set $\mathbf{I}$ as described in Figure 1, with perturbation magnitudes $e = 5/255$ and perturbation dimension $r = 924 \times 1$ and $r = 5544 \times 1$, respectively.

Table 3: This table reports the bound ratio, bound_ratio, and the empirical miscoverage level of our reachset, $\hat{\epsilon}$ for a selection of test images, perturbation level and perturbation dimensions. We performed $10^6$ inferences to estimate the conservatism.

| Dataset | Model | Image name | # Perturbed Pixels | $e$ | bound_ratio | $\hat{\epsilon}$ |
|---------|-------|------------|-------------------|-----|-------------|-------------------|
| Camvid | BiSeNet | 0001TP_008790.png | $41471 \times 3$ | $5/255$ | 0.5328 | 3.08e-6 |

the high perturbation levels considered, both methods encountered out-of-memory errors and failed to complete verification on the same hardware used for our approach. This highlights that while deterministic guarantees are often preferable, they may not always be computationally practical. In such cases, our probabilistic verification method for SSNs offers a scalable and effective alternative.

