# OpenReview forum: "Probabilistic Robustness Analysis in High Dimensional Space: Application to Semantic Segmentation Networks"
_ICLR.cc/2026/Conference — Submitted to ICLR 2026_

### Official Review · Reviewer_RAZP · 2025-10-22

**Soundness:** 2
**Presentation:** 2
**Contribution:** 1
**Rating:** 2
**Confidence:** 3

**Summary:**

Semantic segmentation networks (SSNs) are important in safety-critical domains such as medical imaging and autonomated driving. Existing probabilistic verification techniques struggle to scale to the high complexity and dimensionality of modern SSNs, often resulting in guarantees that are overly conservative and of limited practical use. The authors introduce a scalable and architecture-independent probabilistic verification framework based on conformal inference (CI). They integrate a novel clipping block mechanism that refines uncertainty calibration while preserving formal guarantees. Experiments on large-scale segmentation benchmarks - including CamVid, OCTA-500, Lung Segmentation, and Cityscapes - show that the approach provides reliable safety assurances and tighter guarantees than state-of-the-art probabilistic verification methods.

**Strengths:**

- The authors deal with an interesting and important topic.
- Current research in this area is described by the authors.
- It's good that all the preliminaries are explained, which is quite a lot.
- The different used datasets are good.

**Weaknesses:**

- Right at the beginning, terms such as randomized smoothing and conformal inference are simply assumed to be familiar. A brief explanation and, in particular, a distinction between the two would be helpful for understanding. Not so easy to follow if you have no prior knowledge of CI - everything is derived, but little explanation is given as why it works.
- The preliminaries (section 2) and a large part of section 3 (including algorithms 1 and 2) are taken almost one-to-one from the reference Hashemi et al.
- I would like to see more experiments in comparison to the method used by Hashemi et al., as the approach presented by the authors is very similar to this.
- The computing power required for these experiments is already very high. It would be interesting to compare this with randomized smoothing.
- The Cityscapes results in Appendix D are not available.
- The introduction refers directly to Appendix A, where no notation is provided and is therefore difficult to understand.
- I find that the results are hardly described/explained.

**Questions:**

Conformal inference with reachset in the form of a convex hull, the surrogate model, and the PCA approach have all already been done in Hashemi et al.'s paper. What is the novelty of the approach presented here?

---

> ### Author Response · Authors · 2025-11-16
>
> Hashemi et al. (2025) is the first peer-reviewed work extending conformal inference to high-dimensional settings such as SSNs and applying it for verification,
>
> https://neurips.cc/virtual/2025/loc/san-diego/poster/116265
>
> introducing a new guarantee formulation for SSN verification that is fundamentally different from the formulation used in randomized smoothing. Appendix A provides a comparison between the two formulations and highlights the advantages of the conformal-inference–based approach. Because these formulations are structurally incompatible, meaningful comparisons can only be made against Hashemi et al. (2025); comparisons with randomized smoothing are not feasible, as the guarantee notions are different.
>
> We acknowledge that the primary difference between this paper and Hashemi et al. (2025) is the choice of surrogate model. However, we have shown numerically that this modification leads to substantial improvements, enabling verification in settings where Hashemi et al. encountered failures and scalability limitations. The method verifies the same models under more challenging conditions, in which the previous surrogate does not succeed. We believe a fair evaluation focuses on the contribution stemming from these clear performance gains, rather than focusing on the extent of structural changes from the other work.
>
> Below, we respond to the weaknesses and questions you raised.
>
> ---
>
> **Weakness**
>
> 1- Thanks for this comment, we apply this on the revised document.
>
> 2- We have made a conscious effort to avoid any one-to-one reuse, and we have properly cited Hashemi et al. at every relevant part. Nevertheless, we will revise the document to make the overlap clearly distinguishable, as you suggested.
>
> 3- To compare with the surrogate-based technique of Hashemi et al., we would need to reduce the perturbation dimension (i.e., the number of perturbed pixels), because under the current high-dimensional setting their method produces out-of-memory errors. When the perturbation dimension is reduced, Hashemi et al. indeed outperforms our technique, providing nearly identical robustness values with lower runtime. This is expected: they generate conformity scores via inference on a small surrogate model, while our method relies on linear programming.  However, the contribution of this work is scalability, which necessarily requires evaluating high-dimensional image perturbations—precisely the setting in which their surrogate approach fails due to memory limitations (see our response to the second weakness raised by reviewer 6pGL). Under those realistic, high-dimensional conditions, the comparison we present is the most possible meaningful and feasible one.
>
> 4- The guarantee formulation used in randomized-smoothing–based verification applies to a smoothed approximation of the model, whereas our approach provides guarantees on the exact model itself. Because the formulations target fundamentally different objects, there is no meaningful or direct basis for comparison between the two approaches in the context of verification.
>
> 5- We did not run any new Cityscapes experiments in this paper; the material shown corresponds to the naïve approach introduced by Hashemi et al. In the revised version, we will include our own experiment on Cityscapes, compare it against the naïve technique from Hashemi et al., and provide the full results in Appendix D.
>
> 6- Thanks a lot for this comment. We will apply this in the revised document.
>
> 7 - We provided additional explanations in the appendix, but the main paper was constrained by space limitations. Our plan was to expand the discussion of the results on the extra page granted upon acceptance, where we can describe the findings in more detail.
>
> ---
> Questions
>
> 1- We acknowledge that the primary difference between this paper and Hashemi et al. (2025) is the choice of surrogate model. However, we have shown numerically that this modification leads to substantial improvements, enabling verification in settings where Hashemi et al. encountered failures and scalability limitations. The method verifies the same models under more challenging conditions, in which the previous surrogate does not succeed. This replacement was impactful on the results to the extent that it motivated us to submit the work to ICLR.

---

> > ### Comment · Reviewer_RAZP · 2025-11-19
> >
> > Dear authors, thank you for your reply.
> >
> > You have made an effort to avoid one-to-one reuse, but algorithms 1 and 2, as well as Figure 1, are identical, and the entire notation has been adopted, with only minor changes to the wording.
> > Furthermore, I see only minimal novelty if the surrogate model is simply replaced by a proposal that was already made in the Hashemi paper.
> > For these two main reasons, I'm sticking with reject.

---

> > > ### Author Response · Authors · 2025-11-19
> > >
> > > The main reason we chose to submit this to ICLR was the substantial and huge improvement in the paper’s contribution resulting from the surrogate-replacement change.
> > >
> > > We also mentioned in our response, that we will rewrite the text if the contribution of the paper is approved with the reviewers.

---

### Official Review · Reviewer_6pG7 · 2025-10-27

**Soundness:** 2
**Presentation:** 1
**Contribution:** 1
**Rating:** 2
**Confidence:** 3

**Summary:**

The paper is interested in verification of semantic segmentation networks and obtaining coverage guarantees of their outputs. The proposed approach defines a surrogate model by computing the convex hull of outputs obtained from a training set of adversarial data, and then projects new points on the convex hull by solving a linear programming problem. To deal with high dimensional data, the method relies on PCA learned from the training samples to reduce the dimension before computing the convex hull and the projection. The simple surrogate model can then be efficiently verified.

**Strengths:**

- **Originality:** The method is novel in its direct use of a convex hull of logits to define the surrogate reachset. This is a more direct geometric approach that avoids the need for an additional training step (unlike the ReLU network in Hashemi et al. 2025).

**Weaknesses:**

- **Clarity:** The paper suffers from a severe lack of clarity in defining its contribution. The method and explanation heavily rely on and closely mirror Hashemi et al. (2025). The only substantive difference appears to be the replacement of an additional learned ReLU surrogate network with the direct convex hull projection. This strong reliance makes it nearly impossible to evaluate the paper's unique contribution and necessitates a much clearer and more explicit discussion of the differences.
- **Significance:** The paper claims improved scalability and efficiency as key motivations, stating the prior work "could not handle our perturbation dimensions." This is directly contradicted by the results presented in Hashemi et al. (2025) on the same dataset and settings, often with faster runtime. This undermines the paper's core claim about the need for a new, more efficient approach.
- **Quality:** The experimental section is limited and not convincing. The absence of a comparison with the results obtained by the method in Hashemi et al. (2025) (or any other method) makes the presented results impossible to contextualize or evaluate.

**Questions:**

- Could you explicitly clarify the novel technical and conceptual contributions of this paper? Specifically, beyond replacing the ReLU network with the direct convex hull computation, what fundamental differences exist, and why are they necessary for this problem?
- Could you provide a direct, quantitative comparison of the coverage guarantees, runtime, or any other relevant metric, against their results?
- Could the proposed approach be extended to other pixel-wise prediction tasks, e.g. depth estimation?

---

> ### Author Response · Authors · 2025-11-16
> **Reply to comments from Reviewer 6pGL**
>
> Hashemi et al. (2025) is the first peer-reviewed work extending conformal inference to high-dimensional settings such as SSNs and applying it for verification,
>
> https://neurips.cc/virtual/2025/loc/san-diego/poster/116265
>
> introducing a new guarantee formulation for SSN verification that is fundamentally different from the formulation used in randomized smoothing. Appendix A provides a comparison between the two formulations and highlights the advantages of the conformal-inference–based approach. Because these formulations are structurally incompatible, meaningful comparisons can only be made against Hashemi et al. (2025); comparisons with randomized smoothing are not feasible, as the guarantee notions are different.
>
> We acknowledge that the primary difference between this paper and Hashemi et al. (2025) is the choice of surrogate model. However, we have shown numerically that this modification leads to substantial improvements, enabling verification in settings where Hashemi et al. encountered failures and scalability limitations. The method verifies the same models under more challenging conditions, in which the previous surrogate does not succeed. We believe a fair evaluation focuses on the contribution stemming from these clear performance gains, rather than focusing on the extent of structural changes from the other work.
>
> Below, we respond to the weaknesses and questions you raised.
>
> ---
> **Weakness**
>
> 1- **Clarity**: We already outlined the contribution in the final paragraph of the introduction. As stated there, the contribution is to address the scalability limitations of Hashemi et al. (2025), and our experiments explicitly demonstrate this. To make this point unambiguous, we evaluated the same models and datasets used in Hashemi et al. (2025) and showed that our approach successfully verifies cases in which their method runs into out-of-memory failures.
>
> 2- **Significance**: We agree that Hashemi et al. (2025) achieves faster runtime, and this is expected: their approach trains a small ReLU surrogate once and then generates all $m$ conformity scores via forward passes through that small network. In contrast, our approach solves a linear program for *each* conformity score, which is inherently slower. However, our contribution is not runtime—it is scalability. In all experiments in Hashemi et al. (2025) where the small ReLU surrogate is used, the perturbation dimension (i.e., number of perturbed pixels) is at most $102 \times 3$ (for RGB). In our work, we consider full-image perturbations. For CamVid, this corresponds to $960 \times 720 = 691{,}200$ perturbed pixels, which is orders of magnitude larger than 102.
>
> The method in Hashemi et al. (2025) becomes infeasible under such settings due to some issues:
>
> - **Surrogate training becomes impossible at full-image scale.** Their ReLU surrogate must be trained from the input space to the PCA-based latent space. Training a small ReLU model on an input space in $\mathbb{R}^{691{,}200 \times 3}$ cannot reduce the loss or converge meaningfully, while training on $\mathbb{R}^{102 \times 3}$ remains tractable.
>
>  - **Deterministic reachability becomes computationally prohibitive.** Even a small network with 20 ReLU activations becomes infeasible: with a $691{,}200 \times 3$-dimensional input, essentially all ReLUs activate, yielding $2^{20}$ partitions in obtaining the surrogate reachset and leading to out-of-memory failures. In contrast, with a $102 \times 3$-dimensional input, both a smaller surrogate and fewer active ReLUs are realistic, enabling efficient deterministic reachability.
>
> These scalability barriers—not runtime—are the core motivation, and they prevent Hashemi et al. (2025) from handling the perturbation dimensions we consider.
>
> 3- **Quality**: As mentioned earlier, we agree that on lower-dimensional image perturbations Hashemi et al. (2025) outperforms our technique. However, the main contribution of our work is scalability, and our experiments clearly demonstrate that we can verify models in settings where the surrogate-based approach in Hashemi et al. (2025) cannot operate at all due to its dimensional limitations. Isn't it the most clear possible comparison? Furthermore, comparisons with methods based on randomized smoothing are not feasible because their guarantee formulation is fundamentally different (they verify a smooth approximation rather than the exact model) leaving no valid basis for direct comparison.
>
> ---
> **Questions**
>
> 1- The only difference is the replacement of the surrogate models. However, this replacement was impactful on the results to the extent that it motivated us to confidently submit the work to ICLR.
>
> 2- Their technique encountered out-of-memory errors on settings we consider for the verification, so no numerical results were available for comparison.
>
> 3- Yes, it can be applied to depth estimation. The extension is also more straightforward, because unlike the SSNs the depth estimation produces continuous outputs.

---

> > ### Comment · Reviewer_6pG7 · 2025-11-20
> >
> > Thank you for your reply and the clarifications.
> >
> > Your response confirms that the core contribution lies in changing the surrogate model. While this is an interesting aspect, I find its overall scope and novelty to be limited.
> >
> > As previously mentioned in the initial review, the current experiments presented do not sufficiently highlight the difference brought by your proposed change. Furthermore, it's relevant to note that Hashemi et al. themselves mention falling back to the "naive technique" when perturbations are large, meaning they can process large perturbations and do not solely rely on the ReLU network in such cases. This potentially mitigates one of the distinctions you claim.
> >
> > Finally, a major concern that persists (and was also raised by reviewer RAZP) is the almost identical structure, descriptions, figures, and algorithms to those presented in Hashemi et al. (2025).

---

> > > ### Author Response · Authors · 2025-11-20
> > >
> > > Thanks for your reply
> > >
> > > We can rewrite the text totally if the contribution of the paper is approved and is acceptable with the reviewers.

---

> > > > ### Author Response · Authors · 2025-11-21
> > > >
> > > > We also emphasize that the key contribution here is avoiding a fallback to the naive method when the ReLU-surrogate approach becomes impractical. The naive method is inherently more conservative, and Appendix C demonstrates that our new technique both reduces this conservatism and avoids the scalability limitations that motivates the fallback in the first place in Hashemi et al.

---

### Official Review · Reviewer_SLRv · 2025-10-31

**Soundness:** 3
**Presentation:** 2
**Contribution:** 3
**Rating:** 6
**Confidence:** 3

**Summary:**

In this work, the authors propose a probabilistic verification framework for semantic segmentation networks (SSNs) using conformal inference enhanced with a novel clipping block. The method addresses limitations of existing work by replacing a trained ReLU surrogate model with a convex hull projection approach. They conduct experiments on 4 large-scale segmentation datasets, namely, CamVid, OCTA-500, Lung segmentation, and Cityscapes and demonstrate scalability to perturbations.

**Strengths:**

- The clipping block is training-free which is a big advantage compared. Hence it can be used in a plug-and-play manner with any existing model.
- The authors provide extensive formal proofs and guarantees regarding probabilistic coverage.
- Extensive experiments (4 large and popular segmentation datasets, several perturbation dimensions and magnitudes)

**Weaknesses:**

- The use of PCA and linear programming to project onto a convex hull in high dimensions is computationally expensive
- Authors do not provide any study of sensitivity to N (parameter in PCA)
- Authors should compare to other baselines such as randomized smoothing, hashemi etc, atleast on the toy example
- Authors should provide analysis of how changing confidence levels ($\delta_1, \delta_2$) affects tightness of robustness
- Currently the authors focus on norm perturbations but it would be interesting if authors could provide a discussion on other perturbations like blur, affine transformations, brightness etc
- The authors only use $l \infty$ but it would be interesting to compare $l_1,l_2$ too

**Questions:**

Please see weakness section

---

> ### Author Response · Authors · 2025-11-17
> **Reply to reviewer SLRv**
>
> Thanks for the fair assessment of this work. Unlike the other reviewers who were primarily focusing on structural differences from Hashemi et al. (2025), you focused on the actual contribution relative to that prior work, and we genuinely appreciate that. In many cases, a single change can yield substantial benefits and resolve significant limitations of an earlier technique, and that is exactly what our contribution achieves.
>
> Below, we respond to the weaknesses and questions you raised.
>
> ---
> **Weakness**
>
> 1- The PCA technique we use is a deflation-based variant trained with stochastic gradient descent. It estimates the top principal directions in order of importance and scales well to high-dimensional settings. In addition, the linear programming–based projection is applied only after the data has been reduced into the latent space, where the dimensionality is much smaller. This makes the computation tractable, and our numerical results confirm that it is feasible in practice.
>
> 2- Thank you for this comment. We will run a new ablation study on the PCA parameter $N$, as well as on the calibration-set size $m$, and include the results in the revised version of the paper.
>
>  3- The randomized-smoothing–based methods verify a guarantee formulation that is fundamentally different from the one used in Hashemi et al., and they verify a smoothed approximation of the model rather than the exact model. These differences make a direct, meaningful comparison with randomized smoothing impossible. Moreover, randomized smoothing does not rely on reachability analysis, which is the focus of the toy example in Appendix C. However, we agree that the comparison in Appendix C is not fully complete. We only compared against the naïve technique from Hashemi et al., since our goal there was to illustrate the improvement of our approach relative to that baseline. We did not include the surrogate-based technique from Hashemi et al. on this toy example, and we will add its reachable set to Appendix C in the revised version to provide a more comprehensive comparison.
>
> 4- Because our verification is probabilistic, the robustness value we report is also probabilistic. For example, stating RV = 99.72 is not a complete statement unless the associated confidence levels $\delta_1$ and $\delta_2$ are also specified. The tuple $(\text{RV}, \delta_1, \delta_2)$ together forms the full guarantee:
>
>
> $\Pr\big[ \Pr[\text{RV} = 99.72] > \delta_1 \big] > \delta_2.$
>
> Reporting RV alone is therefore incomplete. The “tightness” of robustness corresponds to the conservatism of the verification:
>
> * higher conservatism → lower tightness,
> * lower conservatism → higher tightness.
>
> Importantly, the conservatism in our method is **not** driven by $(\delta_1, \delta_2)$. Instead, it is determined by:
>
> * **Fidelity of the surrogate model:** A more accurate surrogate reduces the approximation error to the real model, leading to a smaller quantile for conformity scores (driven by approximation errors), and thus a smaller inflated reachset, and therefore a tighter robustness estimate.
>
> * **Size of the calibration dataset $m$:** Increasing $m$ allows the quantile of conformity scores to be estimated less conservatively via conformal inference. A smaller quantile reduces the inflation applied to the surrogate reachset, which yields a smaller reachable set and a tighter robustness bound.
>
> 5- We have already demonstrated that our technique has no scalability limitations when verifying full-image norm perturbations. Because the core steps of our approach—PCA-based dimensionality reduction and convex-hull projection in the latent space—do not rely on the perturbation being an $\ell_p$-norm ball specifically, the method can naturally accommodate other pixel-wise or image-level transformations such as blur, affine transformations, and brightness adjustments. These perturbations can be expressed as transformations applied to the input before projection into the latent space, and the scalability characteristics remain unchanged. We will include experiments for these additional perturbation types in the revised document.
>
>
> 6 - For input-space perturbations, we did consider both $\ell_\infty$ and $\ell_2$ in the numerical results when performing verification. This is while our verification approach can handle all types of image full-perturbations. However, for the projection step in the output space (i.e., the clipping block), we restricted ourselves to $\ell_\infty$. Our experimental observations showed that using $\ell_2$ for the projection requires quadratic programming and was significantly more memory-intensive and introduced major efficiency issues, while $\ell_1$ projection was also substantially slower than $\ell_\infty$. For these reasons, we designed the toolbox around the $\ell_\infty$ projection, as it offered the most practical and scalable performance without sacrificing the validity of the guarantees.

---

### Official Review · Reviewer_LpMV · 2025-11-01

**Soundness:** 2
**Presentation:** 2
**Contribution:** 2
**Rating:** 2
**Confidence:** 3

**Summary:**

This paper proposes a new probabilistic verification framework for semantic segmentation networks (SSNs). Existing methods for certified robustness (especially randomized smoothing) struggle to scale to high-dimensional segmentation tasks. The authors propose a scalable, architecture-agnostic approach combining conformal inference (CI) with a novel clipping block surrogate model. The clipping block projects network outputs onto a convex hull formed from training logits, avoiding the need for training a separate surrogate network (as in Hashemi et al., 2025). The approach provides provable $(\epsilon, l, m)$ probabilistic guarantees, supports general $L_p$ perturbations, and reduces conservatism. Experiments on CamVid, OCTA-500, Lung Segmentation, and Cityscapes demonstrate improved scalability and less conservative robustness bounds compared to prior CI and randomized smoothing methods.

**Strengths:**

- The “clipping block” is an elegant, training-free replacement for surrogate ReLU networks, avoiding fidelity and scalability issues.
- The paper extends CI to large-scale probabilistic reachability with formal $(\epsilon, l, m)$ guarantees.
- Demonstrated on realistic, high-dimensional datasets and large segmentation models (UNet, BiSeNet, HRNetV2).
- The paper systematically contrasts its approach with prior CI and randomized smoothing methods.
- Includes an anonymous toolbox and detailed algorithmic pseudocode.

**Weaknesses:**

- While multiple datasets are used, the evaluation focuses on a narrow type of perturbation (darkening) and may not reflect broader robustness (e.g., geometric or semantic transformations).
- The empirical section omits comparisons to recent certified robustness methods beyond Hashemi et al. (2025) and smoothing approaches.
- The convex hull projection step is computationally heavy; PCA-based dimensionality reduction mitigates this but may introduce approximation bias.
- The paper’s exposition is dense and overly mathematical in sections 3,4, which may obscure intuition for readers less familiar with CI.
- No ablation on PCA vs. clipping: It’s unclear how much each contributes to scalability and accuracy improvements.

**Questions:**

- How sensitive is the robustness value (RV) to the choice of calibration size $m$ and the PCA dimensionality $N$?
- Can the convex hull projection scale beyond the datasets tested?
- How would this approach handle distributional shift in test data; does the CI guarantee still hold under covariate drift?
- Could the clipping block approach be combined with randomized smoothing to strengthen guarantees?
- What is the empirical runtime or memory bottleneck for convex hull construction as t and n grow?
- Is it possible to simplify the presentation for the reader, and add additional retails (relevant background etc.) in the Appendix?

---

> ### Author Response · Authors · 2025-11-14
> **Response to the comments**
>
> *Hashemi et al. (2025)* is the first peer-reviewed work applying conformal inference to SSNs,
>
> https://neurips.cc/virtual/2025/loc/san-diego/poster/116265
>
> introducing a guarantee formulation fundamentally different from randomized smoothing. Because of this difference in formulation, meaningful comparisons can only be made against that work. Our submission replaces one key surrogate component, and this modification resolves the limitations reported in *Hashemi et al. (2025)*. The method verifies the same models under more challenging conditions, where the previous surrogate fails. Thus, the contribution lies in the clear performance improvement rather than the magnitude of structural changes relative to the prior work.
>
> Below, we respond to the weaknesses and questions you raised.
>
> ---
>
> **Weakness:**
>
> 1- The evaluation does also include full-image perturbations. As shown in Figure 2, the first row reports verification results for $ \ell_2 $ and $ \ell_\infty $ perturbations applied over the entire image. Since the method successfully verifies these full-image attacks, extending the evaluation to additional perturbation types (e.g., geometric or semantic transformations) is feasible, and we will incorporate such cases in the final version of the paper.
>
> 2- SSNs present a fundamentally different challenge due to their high-dimensional input-output space. To date, the only probabilistic verification methods developed specifically for SSNs are randomized smoothing–based approaches and the recent work of *Hashemi et al. (2025)*. As a result, there are no additional SSN-specific probabilistic verification methods available in the literature to compare against.
>
> 3- Exactly. Our method does not produce an exact reach set; it yields a valid but conservative approximation. This conservatism is the necessary cost for making verification feasible and scalable on high-dimensional SSNs. However, the numerical results in Figure 2 indicate that this conservatism is not overly limiting. For example, for large-scale models such as BiseNet, UNet1, and UNet2 (see Table 1), we obtain robustness estimates as high as $ \bar{RV} = 99.72% $. on BiseNet, which suggests that the approximation bias introduced by the surrogate does not materially undermine the resulting guarantees.
>
> 4,5- Yes, we will consider these comments in the revised version of the text.
>
> ---
>
> **Questions**
>
> ---
>
> 1- The sensitivity of the robustness value ($RV$) to the calibration size ($m$) and the PCA dimensionality ($N$) depends strongly on the underlying model. In many cases, the model is sufficiently robust that even a conservative reach set does not change the number of pixels classified as robust. In general, however, the following principles apply:
>
> • Effect of $N$: Reducing $N$ relative to the original dimension $n$ increases the approximation error introduced by PCA. This results in a larger inflation of the convex hull $CH$, producing a more conservative (and larger) reachable set. A larger reachable set is more likely to classify some truly robust pixels as *unknown*, thereby reducing the RV.
>
> • Effect of $m$: When the confidence levels $(\delta_1, \delta_2)$ are fixed, increasing $m$ reduces the conservatism of the conformal inference step. This leads to a smaller inflation factor, which directly reduces the amount by which the convex hull $CH$ is inflated. With a less-inflated $CH$, pixels that were previously marked as *unknown* (due to the larger inflation when $m$ was smaller) can now be certified as robust.
>
> 2- The key point is that the guarantee itself remains valid even when the surrogate model introduces significant approximation error. This means that for higher-dimensional datasets, we can always reduce $N$ to a level where the LP remains scalable and then apply the convex-hull projection. Doing so will increase conservatism, but it does not invalidate the guarantee; the method continues to provide correct (though potentially more conservative) robustness certificates.
>
> 3- We talked about this in Appendix A. The paper [ https://arxiv.org/abs/2008.04267 ] demonstrates that conformal inference can be made robust to distribution shift with very low additional cost. Their approach suggests adjusting the rank used when sorting residuals—essentially selecting a more conservative quantile—to maintain valid guarantees under covariate drift. This modification is simple to apply and provides robustness to distributional shift without materially increasing computation.
>
> 4- The main contribution of clipping block is to provide a deterministic reachset while preserving an acceptable approximation error with SSN. These benefits are only beneficial when we verify through reachability. The randomized smoothing approach does not use reachability.
>
> 5- The runtime is more dependent on inference over SSN, but increasing $t$ and $n$ provides issues for saving the convex hull on the GPU or CPU RAM.
>
> 6- Yes, we consider this in the revised version of the text.

---

> > ### Comment · Reviewer_LpMV · 2025-11-28
> >
> > I'd like to thank the authors for their response, which clarifies some of my questions. However, the limited novelty in comparison with Hashemi et al. (2025), and the significant overlap with that prior work are important concerns - I'm not sure if I can raise my score at this time.. In the case of same authors / lab, would maybe a journal publication be more appropriate?

---

> ### Author Response · Authors · 2025-11-28
>
> Thanks for your reply.
>
> We introduce a more suitable surrogate model for that verification algorithm, replace their surrogate with ours, and show that this substitution resolves all the previously reported scalability issues. Although the difference lies in the choice of surrogate models, a single change can often yield substantial benefits and eliminate major limitations of an earlier technique, and that is exactly what our contribution achieves.
>
> We intentionally verified the same SSNs on the same dataset as Hashemi et al., but with more difficult settings, to show that our surrogate enables verification in settings that are not feasible when their surrogate is used. This demonstrates that we provide the needed scalability for that technique, and this was our core contribution, convincingly supported by the numerical results. We also observed that their surrogate model consistently resulted in out-of-memory errors for the settings considered in our experiments.
>
> The contribution was substantial to the point that we confidently decided to submit this work to ICLR.

---

### Meta-Review · Area_Chair_ruAz · 2026-01-08

**Summary:**

A key concern of the reviewers was the lack of novelty, compared to Hashemi et al (2025). The paper suggests making a substitution in the existing framework, that was alluded to in the original paper. A significant majority of the reviewers agreed that this was not sufficient for publication.

**Reviewer Concerns:**

Several concerns could not be addressed without a further review cycle, including the novelty, and limited type of perturbations. While the authors state that different perturbations could be used, this was not demonstrated in the current paper.

**Reviewer Scores:**

All reviewers did engage with the rebuttal from the authors, and so little additional changes seem likely.

---

### Decision · Program_Chairs · 2026-01-26

Reject